# RISK CONTROL FOR ONLINE LEARNING MODELS

## ABSTRACT

To provide rigorous uncertainty quantification for online learning models, we develop a framework for constructing uncertainty sets that provably control risk—such as coverage of confidence intervals, false negative rate, or F1 score—in the online setting. This extends conformal prediction to apply to a larger class of online learning problems. Our method guarantees risk control at any user-specified level even when the underlying data distribution shifts drastically, even adversarially, over time in an unknown fashion. The technique we propose is highly flexible as it can be applied with any base online learning algorithm (e.g., a deep neural network trained online), requiring minimal implementation effort and essentially zero additional computational cost. We further extend our approach to control multiple risks simultaneously, so the prediction sets we generate are valid for all given risks. To demonstrate the utility of our method, we conduct experiments on real-world tabular time-series data sets showing that the proposed method rigorously controls various natural risks. Furthermore, we show how to construct valid intervals for an online image-depth estimation problem that previous sequential calibration schemes cannot handle.

## 1 INTRODUCTION

To confidently deploy learning models in high-stakes applications, we need both high predictive accuracy and reliable safeguards to handle unanticipated changes in the underlying data-generating process. Reasonable accuracy on a fixed validation set is not enough, as raised by Sullivan (2015); we must also quantify uncertainty to correctly handle hard input points and take into account shifting distributions. For example, consider the application of autonomous driving, where we have a real-time view of the surroundings of the car. To successfully operate such an autonomous system, we should measure the distance between the car and close-by objects, e.g., via a sensor that outputs a depth image whose pixels represent the distance of the objects in the scene from the camera. Figure 1a displays a colored image of a road and Figure 1b presents its corresponding depth map. Since high-resolution depth measurements often require longer acquisition time compared to capturing a colored image, there were developed online estimation models to predict the depth map from a given RGB image (Patil et al., 2020; Zhang et al., 2020). The goal of these methods is to artificially speed-up depth sensing acquisition time. However, making decisions solely based on an estimate of the depth map is insufficient as the predictive model may not be accurate enough. Furthermore, the distribution can vary greatly and drastically over time, rendering the online model to output highly inaccurate and unreliable predictions. In these situations, it is necessary to design a predictive system that reflects the range of plausible outcomes, reporting the uncertainty in the prediction. To this end, we encode uncertainty in a rigorous manner via prediction intervals/sets that augment point predictions and have a long-range error control. In the autonomous driving example, the uncertainty in the depth map estimate is represented by depth-valued uncertainty intervals. In this paper, we introduce a novel calibration framework that can wrap any online learning algorithm (e.g., an LSTM model trained online) to construct prediction sets with guaranteed validity.

Formally, suppose an online learning setting where we are given data stream $\{(X_t, Y_t)\}_{t \in \mathbb{N}}$ in a sequential fashion, where $X_t \in \mathcal{X}$ is a feature vector and $Y_t \in \mathcal{Y}$ is a target variable. In single-output regression settings $\mathcal{Y} = \mathbb{R}$, while in classification tasks $\mathcal{Y}$ is a finite set of all class labels. The input $X_t$ is commonly a feature vector, i.e., $\mathcal{X} = \mathbb{R}^p$, although it may take different forms, as in the depth sensing task, where $X_t \in \mathbb{R}^{M \times N \times 3}$ is an RGB image and $Y_t \in \mathbb{R}^{M \times N}$ is the ground truth depth. Consider a loss function $L(Y_t, \hat{C}_t(X_t)) \in \mathbb{R}$ that measures the error of the estimated prediction

set $\hat{C}_t(X_t) \subseteq \mathcal{Y}$ with respect to the true outcome $Y_t$. Importantly, at each time step $t \in \mathbb{N}$, given all samples previously observed $\{(X_i, Y_i)\}_{i=1}^{t-1}$ along with the test feature vector $X_t$, our goal is to construct a prediction set $\hat{C}_t(X_t)$ guaranteed to attain any user-specified risk level $r$:

$$\mathcal{R}(\hat{C}) = \lim_{T \to \infty} \frac{1}{T} \sum_{t=1}^{T} L(Y_t, \hat{C}_t(X_t)) = r. \tag{1}$$

For instance, a natural choice for the loss $L$ in the depth sensing task is the image miscoverage loss:

$$L_{\text{image miscoverage}}(Y_t, \hat{C}(X_t)) = \frac{1}{MN} \left| (m, n) : Y_t^{m,n} \notin \hat{C}^{m,n}(X_t) \right|. \tag{2}$$

In words, $L_{\text{image miscoverage}}(Y_t, C(X_t))$ is the ratio of pixels that were miscovered by the intervals $\hat{C}^{m,n}(X_t)$, where $(m, n)$ is the pixel's location. Hence, the resulting risk for the loss in (2) measures the average image miscoverage rate across the prediction sets $\{\hat{C}_t(X_t)\}_{t=0}^{\infty}$, and $r = 20\%$ is a possible choice for the desired miscoverage frequency. Another example of a loss function that is attractive in multi-label classification problems is the false negative proportion whose corresponding risk is the false negative rate.

In this work, we introduce *rolling risk control* (`Rolling RC`): the first calibration procedure to form prediction sets in online settings that achieve any pre-specified risk level in the sense of (1) without making any assumptions on the data distribution, as guaranteed by Theorem 1. We accomplish this by utilizing the mathematical foundations of *adaptive conformal inference* (`ACI`) (Gibbs & Candes, 2021) which is a groundbreaking conformal calibration scheme that constructs prediction sets for any arbitrary time-varying data distribution. The uncertainty sets generated by `ACI` are guaranteed to have valid long-range coverage, being a special case of (1) with the choice of the 0-1 loss (indicator function) defined in Section 2. Importantly, one cannot simply plug an arbitrary loss function into `ACI` and achieve risk control. The reason is that `ACI` works with conformity scores—a measure of goodness-of-fit—that are only relevant to the 0-1 loss, but do not exist in the general risk-controlling setting. Therefore, our `Rolling RC` broadens the set of problems that `ACI` can tackle, allowing the analyst to control an arbitrary loss. Furthermore, the technique we proposed in Section 3.3 is guaranteed to control multiple risks, and thus constructs sets that are valid for all given risks over long-range windows in time. Additionally, the proposed online calibration scheme is lightweight and can be integrated with any online learning model, with essentially zero added complexity. Lastly, in Section 3.2.1 we carefully investigated design choices of our method to adapt quickly to distributional shifts. Indeed, the experiments conducted on real benchmark data sets, presented in Section 4, demonstrate that sophisticated designed choices lead to improved performance.

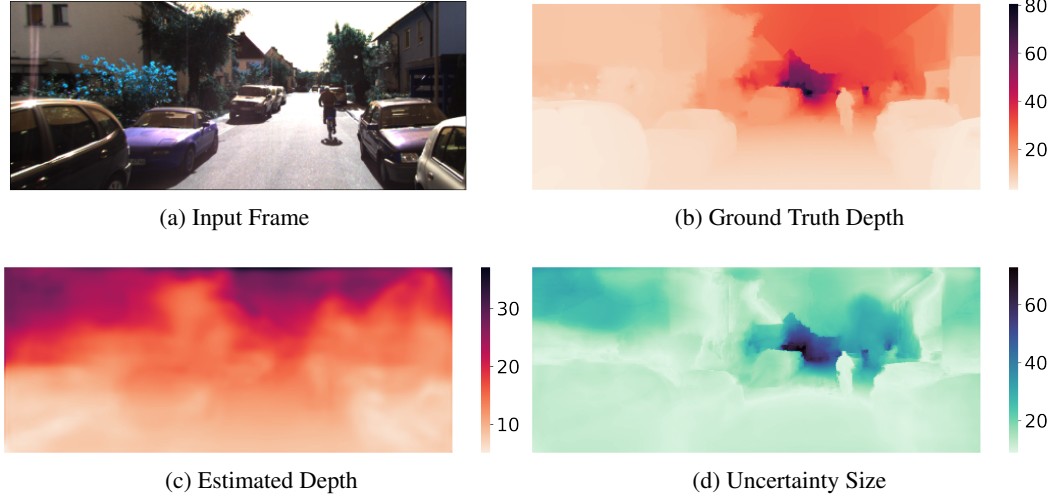

(a) Input Frame        (b) Ground Truth Depth

(c) Estimated Depth        (d) Uncertainty Size

Figure 1: Online depth estimation. The input frame, ground truth depth map, estimated depth image, and interval's size at time step $t = 8020$. All values are in meter units.

## 1.1 Uncertainty Quantification for Online Depth Estimation

Recall the online depth sensing problem, where our goal is to construct a prediction interval $C^{m,n}(X_t) \subset \mathbb{R}$ for each pixel $(m, n)$ that contains the ground truth depth $Y_t^{m,n}$ at 80% frequency. That is, we aim at controlling the image miscoverage loss (2) at level $r = 20\%$. To accomplish this, we apply our `Rolling RC` framework using a neural network model for depth estimation; in Appendix C.2.4 we give more information regarding the online training scheme and the implementation details. At a high level, we fit (offline) an initial predictive model on the first 6000 samples to obtain a reasonable predictive system. Next, passing time step 6001, we proceed by training the model in an online fashion while applying our calibration procedure, and then measure the performance on the data points corresponding to time steps 8001 to 10000.

Figure 1c shows the estimated depth image generated by the base model and Figure 1d displays the size of the prediction interval of each pixel at timestamp $t = 8020$. These figures suggest that the calibrated uncertainty intervals properly reflect the ground truth depth. Furthermore, Figure 2 presents the image coverage rate and average length across the test timestamps, revealing that the proposed method accurately controls the risk with an average image coverage rate of 80%.

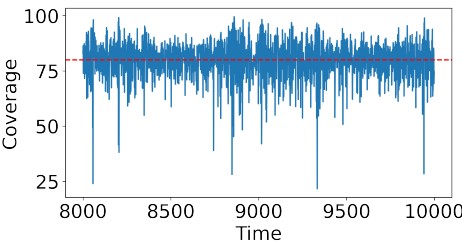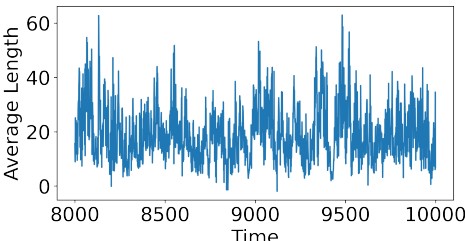

Figure 2: The coverage rate and average interval length (in meters) over each image in the test sequence achieved by the proposed uncertainty quantification method. The average coverage is 80.00% and the average length is 18.04 meters.

## 2 Background

Conformal inference (Angelopoulos & Bates, 2021; Vovk et al., 2005) is a generic approach for constructing prediction sets in regression or classification tasks that attain any pre-specified coverage rate, under the assumption that the training and testing points are i.i.d., or exchangeable. One example of such a method is *Split conformal prediction* (Lei et al., 2018; Papadopoulos et al., 2008). In a nutshell, the idea is to split the observed labeled data into training and calibration sets, fit a model on the training set, and evaluate the model's goodness-of-fit on the reserved holdout calibration points. Under the i.i.d assumption stated above, the prediction sets produced by split conformal prediction are guaranteed to have the following coverage property: $1 - \mathbb{E}[\mathbb{1}\{Y \notin C(X)\}] \geq 1 - \alpha$, where $(X, Y)$ is a fresh data point and $\alpha$ is a pre-specified miscoverage rate.

While coverage rate is an important property, in real-world applications, it is often desired to control metrics other than the binary loss $\mathbb{1}\{Y \notin C(X)\}$ that defines the coverage requirement. Such losses include the F1-score or the false negative rate, where the latter is attractive for data with high-dimensional $Y$ as in image segmentation tasks or image recovery applications. Indeed, there have been developed extensions of the conformal approach that go beyond the 0-1 loss, rigorously controlling more general loss functions (Angelopoulos et al., 2021a; 2022a; Bates et al., 2021). Analogously to split conformal prediction, these methods provide a *risk-controlling* guarantee that holds under the i.i.d. assumption. In particular, such guarantees do not hold for time-varying data with arbitrary distributional shifts, as the i.i.d. assumption would not hold anymore.

Since the i.i.d assumption of the conformal approach is often violated in real-world applications, there have been developed extensions to conformal inference that impose relaxed notions of exchangeability (Cauchois et al., 2020; Chernozhukov et al., 2018; Stankeviciute et al., 2021; Tibshirani et al., 2019; Xu & Xie, 2021a;b). Such methods, however, are not guaranteed to construct prediction sets with a valid coverage rate for general time-series data with arbitrary distributional shifts. By contrast, `ACI`

(Gibbs & Candes, 2021) generates uncertainty sets that are guaranteed to achieve a user-specified level of the 0-1 loss:

$$L_{0\text{-}1}(Y_t, C(X_t)) = \mathbb{1}_{\{Y_t \notin C_t(X_t)\}} = \begin{cases} 1, & Y_t \notin C_t(X_t), \\ 0, & \text{otherwise.} \end{cases} \tag{3}$$

A recent work by Gibbs & Candès (2022) proposes a more sophisticated approach to track past coverage rates to better adapt to distributional shifts. In general, this line of research is based on a common, simple idea—if the past coverage rate is too high, we shorten the intervals, and if it is too low, we widen them. In this paper, we also rely on the above update rule, however, guarantee the control of a general risk, standing in contrast with `ACI` that controls only the binary loss in (3). Therefore, our approach is the first online calibration scheme that can control risks other than the coverage frequency.

## 3 PROPOSED METHOD

### 3.1 GENERAL FORMULATION

We now turn to present `Rolling RC`—a general framework for uncertainty quantification in an online learning setting, which satisfies the risk requirement in (1). Towards that end, we define a set construction function

$$\hat{C}_t(X_t) = f(X_t, \theta_t, \mathcal{M}_t) \in 2^{\mathcal{Y}} \tag{4}$$

that gets as an input (i) the test $X_t$, (ii) a fitted model $\mathcal{M}_t$ trained on all data $\{X_{t'}, Y_{t'}\}_{t'=1}^{t-1}$ up to time $t$, and (iii) a calibration parameter $\theta_t$, and returns a prediction set. Above, $2^{\mathcal{Y}}$ is the power set of $\mathcal{Y}$. For instance, in the depth prediction task in Section 1.1, $f$ constructs a prediction interval for each pixel in the image, as visualized in Figure 1d. The model $\mathcal{M}_t(X_t)$ is used to form a prediction for $Y_t$ given the current feature vector $X_t$; we will provide soon concrete formulations for the set constructing function $f$ as well as examples for $\mathcal{M}$. The calibration parameter $\theta_t \in \mathbb{R}$ controls the size of the prediction set generated by $f$: larger $\theta_t$ leads to larger sets, and smaller $\theta_t$ leads to smaller sets. Under the assumption that larger sets produce a lower loss, $\theta_t$ allows us to control the risk over long-range windows in time: by increasing (resp. decreasing) $\theta_t$ over time we increase (resp. decrease) the empirical risk. Once $Y_t$ is revealed to us, we tune $\theta_t$ according to the following rule:

$$\theta_{t+1} = \theta_t + \gamma(l_t - r). \tag{5}$$

This update rule is exactly that of `ACI` (Gibbs & Candès, 2022), extended to our more general setting. Above, $l_t = L(Y_t, C(X_t))$ is the loss at time $t$, and $\gamma > 0$ is a fixed step size, e.g., 0.05. In Appendix C.3.3 we study the effect of $\gamma$ on the resulted sets and provide a suggestion for properly setting it. The pre-defined constant $r$ is the desired risk level, specified by the user, e.g., 0.2 for the image miscoverage loss in (2). Lastly, we obtain a new predictive model $\mathcal{M}_{t+1}$ by updating the previous $\mathcal{M}_t$ with the new labeled pair $(X_t, Y_t)$, e.g., by applying a single gradient step to reduce any given predictive loss function. For convenience, the `Rolling RC` procedure is summarized in Algorithm 1. The validity of our proposal is given below, whose proof is deferred to Appendix A.1. In Appendix A.2 we introduce a more general theorem that extends the domain of $\hat{C}_t$ beyond the power set $2^{\mathcal{Y}}$.

**Theorem 1.** *Suppose that $f : (\mathcal{X}, \mathbb{R}, \mathbb{M}) \to 2^{\mathcal{Y}}$ is an interval/set constructing function. In addition, suppose that there exist constants $m$ and $M$ such that for all $x \in \mathcal{X}$, $y \in \mathcal{Y}$ and $\mathcal{M} \in \mathbb{M}$, $f(x, \theta, \mathcal{M}) = \mathcal{Y}$ for all $\theta > M$, $f(X, \theta, \mathcal{M}) = \emptyset$ for all $\theta < m$. Further suppose that the loss is bounded and satisfies $L(y, \mathcal{Y}) < r$ and $L(y, \emptyset) > r$. Consider the following series of calibrated intervals: $\{\hat{C}_t(X_t)\}_{t=1}^{\infty}$, where $\hat{C}_t(X_t)$ is defined according to (4). Then, the calibrated intervals satisfy the risk requirement in (1).*

Crucially, this theorem states that the risk-control guarantee of `Rolling RC` holds for any distribution $\{P_{X_t, Y_t}\}_t$, any set-valued function $f$, and any sequence of online-updated predictive models $\{\mathcal{M}_t\}_t$. The requirements for Theorem 1 to hold are (i) the function $f$ must yield the empty set for small enough $\theta$ and the full label space for large enough $\theta$; and (ii) the loss is smaller than the desired level $r$ for the full label set $\mathcal{Y}$ and exceeds $r$ for the empty set. Also, note that the step size $\gamma$ and the bounds $m, M$ must be fixed in order to control the risk.

---

**Algorithm 1** `Rolling RC`

---

**Input:**

Data $\{(X_t, Y_t)\}_{t=1}^T \subseteq \mathcal{X} \times \mathcal{Y}$, given as a stream, desired risk level $r \in \mathbb{R}$, a step size $\gamma > 0$, a set constructing function $f : (\mathcal{X}, \mathbb{R}, \mathbb{M}) \to 2^{\mathcal{Y}}$ and an online learning model $\mathcal{M}$.

**Process:**

1: Initialize $\theta_0 = 0$.
2: **for** $t = 1, ..., T$ **do**
3:     Construct a prediction set for the new point $X_t$: $\hat{C}_t(X_t) = f(X_t, \theta_t, \mathcal{M}_t)$.
4:     Obtain $Y_t$.
5:     Compute $l_t = L(Y_t, \hat{C}_t(X_t))$.
6:     Update $\theta_{t+1} = \theta_t + \gamma(l_t - r)$.
7:     Fit the model $\mathcal{M}_t$ on $(X_t, Y_t)$ and obtain the updated model $\mathcal{M}_{t+1}$.
8: **end for**

**Output:**

Uncertainty sets $\hat{C}_t(X_t)$ for each time step $t \in \{1, ...T\}$.

---

Observe that our method is immune to over-fitting by design even though we use the same data point twice: the evaluation of $\theta_t$ is conducted by using the predictions produced by an "old" model, before updating it with the new labeled point. We also note that the proof of Theorem 1 gives finite-sample bounds for how close the realized risk is to the desired level; the empirical risk falls within a $C/T$ factor of the desired level $r$, where $C = (M - m + 4 \cdot \gamma B)$ is a known constant; see Appendix A.1 for details. Furthermore, our new formulation unlocks new design possibilities: we can define any prediction set function $f$ while guaranteeing the validity of its output by calibrating any parameter $\theta_t$. This parameter can affect $f$ in a highly non-linear fashion to attain the most informative (i.e., small) prediction sets. Of course, the performance of the proposed scheme is influenced by the design of $f$, which is the primary focus of the next sections.

### 3.2 ROLLING RC FOR REGRESSION

In this section, we focus on 1-dimensional response variable $Y$ and aim to control the 0-1 loss in (3). Note that in Section 4.1.3 we will also deal with 1-dimensional responses but show how to control a more sophisticated notion of error for which the loss is defined on the time-horizon, and in Appendix B we provide a concrete scheme for handling a multi-dimensional $Y$.

Suppose we are interested in constructing prediction intervals with $1 - \alpha$ coverage frequency, using a *quantile regression* model $\mathcal{M}_t$ that produces estimates for the $\alpha/2$ and $1 - \alpha/2$ conditional quantiles of the distribution of $Y_t \mid X_t$. We denote these estimates as $\mathcal{M}_t(X_t, \alpha/2)$ and $\mathcal{M}_t(X_t, 1 - \alpha/2)$, respectively, which can be obtained by fitting an LSTM model that minimizes the pinball loss; see Appendix C.1.2 for further details. The guiding principle here is that a model that perfectly estimates the conditional quantiles will form tight intervals with valid risk, attaining $1 - \alpha$ coverage level. In practice, however, the model $\mathcal{M}_t$ may not be accurate, and thus may result in invalid coverage; this is especially true for data with frequent time-varying distributional shifts. Consequently, to ensure valid coverage control, we apply `Rolling RC`. Taking inspiration from the method of conformalized quantile regression (CQR) (Romano et al., 2019), we use the following interval construction function:

$$f(X_t, \theta_t, \mathcal{M}_t) = [\mathcal{M}_t(X_t, \alpha/2) - \varphi(\theta_t), \; \mathcal{M}_t(X_t, 1 - \alpha/2) + \varphi(\theta_t)]. \tag{6}$$

Above, the interval endpoints are obtained by augmenting the lower and upper estimates of the conditional quantiles by an additive calibration term $\varphi(\theta_t)$. The role of $\theta_t$ is the same as before: the larger $\theta_t$, the wider the resulting interval is. Here, however, we introduce an additional stretching function $\varphi$ that can scale $\theta_t$ non-linearly, providing us the ability to adapt more quickly to severe distributional shifts. We now present and review three design options for the stretching function $\varphi$.

### 3.2.1 STRETCHING FUNCTIONS FOR FASTER ADAPTATION

**None.** The first and perhaps most natural choice is $\varphi(x) = x$, which does not stretch the scale of the interval's adjustment factor. While this is the most simple choice, it might be sub-optimal when an aggressive and fast calibration is required. To see this, recall (5), and observe that the step size $\gamma$

used to update $\theta_t$ must be fixed throughout the entire process. As a result, the calibration parameter $\theta_t$ might be updated too slowly, resulting in an unnecessary delay in the interval's adjustment.

**Exponential.** The exponential stretching function, defined as $\varphi(x) = e^x - 1$ for $x > 0$ and $\varphi(x) = -e^{-x} + 1$ for $x \leq 0$, updates the calibration adjustment factor with an exponential rate: $\varphi'(x) = e^x$, even though the step size for $\theta_t$ is fixed. In other words, it updates $\varphi(\theta_t)$ gently when the calibration is mild ($\varphi(\theta_t)$ is close to 0), and faster as the calibration is more aggressive ($\varphi(\theta_t)$ is away from zero).

**Error adaptive.** The following stretching function updates $\theta_t$ more rapidly when the loss of the previous data point $\ell_{t-1}$ is farther from the desired risk $r$. Furthermore, it makes larger updates when $Y_t$ is far from the interval's boundaries. More formally, denote the CQR non-conformity score (Romano et al., 2019) by

$$s_t = \max\{\mathcal{M}_t(X_t, \alpha/2) - Y_t, Y_t - \mathcal{M}_t(X_t, 1 - \alpha/2)\},$$

which measures the signed distance of $Y_t$ from its closest boundary. Next, define

$$\varphi_t(\theta) = \theta + \lambda_t^{\text{error}}, \text{ where } \lambda_t^{\text{error}} = \text{clip}(\lambda_{t-1}^{\text{error}} - \beta^{\text{score}} \cdot s_{t-1} \cdot \exp\left\{\beta^{\text{loss}} \cdot |\ell_{t-1} - r|\right\}, \beta^{\text{low}}, \beta^{\text{high}}),$$

where $\beta^{\text{loss}}, \beta^{\text{score}}, \beta^{\text{low}}$ and $\beta^{\text{high}}$ are hyperparameters. The clipping function $\text{clip}(x, m, M) = \max\{\min\{x, M\}, m\}$ is applied to restrain the effect of an outlier $Y_t$ that is far from the boundaries.

This discussion above sheds light on the great flexibility of `Rolling RC`: we can accurately find the correct adjustment to the uncertainty set while being adaptive to rapid distributional shifts in the data. Furthermore, Theorem 1 guarantees the risk validity regardless of the choice of the stretching function. In Appendix D.1.1 we propose an additional stretching function and compare all proposed stretching functions. This analysis indicates that the 'error adaptive' stretching is the best choice.

## 3.3 CONTROLLING MULTIPLE RISKS

In this section, we show how to control more than one risk and construct intervals that are valid for all given risks. To motivate the need for such a multiple risks controlling guarantee it may be best to consider the depth estimation example from Section 1.1. Here, we may wish to control not only the coverage of the entire depth image, as in Section 1.1, but also the frequency at which the coverage at the center of the image falls below a certain threshold. This design choice meets reality since the coverage at the center falls below $60\%$ in more than $17\%$ of the time-steps, as presented in Figure 10 in Appendix D.2. This figure also indicates that controlling the center coverage does not control the center failure loss. Concretely, we formulate the center failure loss as:

$$L_{\text{center failure}}(Y_t, C(X_t)) = \mathbb{1}\left\{\frac{1}{|\text{center}|}|(m, n) \in \text{center} : Y_t^{m,n} \in C^{m,n}(X_t)| \leq 60\%\right\}. \quad (7)$$

We define the center of an image as the middlemost 50x50 grid of pixels. Controlling the center failure loss at level $r = 10\%$ ensures that more than $60\%$ of the center will be covered for $90\%$ of the images.

More generally, suppose we are given $k$ arbitrary loss functions $\{L_i\}_{i=1}^k$ and aim to control their corresponding risks, each at level $r^i$. In this setting, the set constructing function $f(\cdot)$ gets as an input the test $X_t$, the fitted model $\mathcal{M}_t$, and a calibration *vector* $\underline{\theta}_t \in \mathbb{R}^k$, and returns a prediction set

$$\hat{C}_t(X_t) = f(X_t, \underline{\theta}_t, \mathcal{M}_t) \in 2^{\mathcal{Y}}. \quad (8)$$

Similarly to the single risk-controlling formulation described in Section 3.1, $\underline{\theta}_t$ controls the size of the generated set: by increasing the coordinates in $\underline{\theta}_t$ we encourage the construction of larger sets with lower risks, and we tune it likewise:

$$\underline{\theta}_{t+1}^i = \underline{\theta}_t^i + \underline{\gamma}^i(l_t^i - r^i),$$

where $l_t^i = L_i(Y_t, \hat{C}_t(X_t))$ and $\underline{\gamma}^i > 0, i = 1, \ldots, L$ is the corresponding step size. We now show that this procedure is guaranteed to produce uncertainty sets with valid risks.

**Theorem 2.** *Suppose that $f : (\mathcal{X}, \mathcal{R}, \mathbb{M}) \to 2^{\mathcal{Y}}$ is an interval/set constructing function. In addition, suppose that there exist constants $\{M^i\}_{i=1}^k$ such that for all $X$ and $\mathcal{M}$, $f(X, \underline{\theta}, \mathcal{M}) = \mathcal{Y}$ if $\underline{\theta}^i > M^i$ for some $i \in \{1, ...k\}$. Further suppose that the losses are bounded and satisfy $L^i(y, \mathcal{Y}) < r^i$ for every $y \in \mathcal{Y}$ and $i \in \{1, ...k\}$. Consider the following series of calibrated intervals: $\{\hat{C}_t(X_t)\}_{t=1}^{\infty}$, where $\hat{C}_t(X_t)$ is defined according to (8). Then, the calibrated intervals attain valid risk:*

$$\forall i \in \{1, ..., k\} \quad \exists D^i \in \mathbb{R} \quad \text{s.t} \quad \frac{1}{T} \sum_{t=1}^{T} L_i(Y_t, \hat{C}_t(X_t)) \leq r^i + \frac{D^i}{T} \xrightarrow[T \to \infty]{} r^i.$$

If we further assume that the risks are more synchronized with each other, we can achieve an exact multiple risks control, as stated next.

**Theorem 3.** *Suppose that $f$ is an interval/set constructing function and $\{L^i\}_{i=1}^k$ are loss functions as in Theorem 2. Further suppose that there exist constants $\{m^i\}_{i=1}^k$ such that $f(X, \underline{\theta}, \mathcal{M}) = \emptyset$ if $\underline{\theta}^i < m^i$ for some $i \in \{1, ...k\}$ and that the losses satisfy $L^i(y, \emptyset) > r^i$ for every $y \in \mathcal{Y}$ and $i \in \{1, ...k\}$. Then, the intervals achieve the exact risk:*

$$\forall i \in \{1, ..., k\} : \mathcal{R}(\hat{C}) = \lim_{T \to \infty} \frac{1}{T} \sum_{t=1}^{T} L_i(Y_t, \hat{C}_t(X_t)) = r^i$$

The proofs of the theoretical results are given in Appendix A. In words, Theorem 2 guarantees the validity of the risks, while Theorem 3 guarantees that all risks are exactly controlled by assuming that during the calibration process there are no two coordinates in $\underline{\theta}_t$ such that the first is too low (requires widening the interval), and the other is too high (requires shrinking the interval). Lastly, we note that in this section we presented one implementation of `Rolling RC` to control multiple risks, although other approaches may be valid as well, e.g., techniques with calibration parameters that are independent of the number of risks.

## 4 EXPERIMENTS

### 4.1 SINGLE RESPONSE: CONTROLLING A SINGLE RISK IN REGRESSION TASKS

In this section, we study the effectiveness of our proposed calibration scheme for time series data with 1-dimensional response variables. Towards that end, we describe two performance metrics that we will use in the following numerical simulations to assess conditional/local coverage rate.

#### 4.1.1 TIME-SERIES CONDITIONAL COVERAGE METRICS

**MC:** The *miscoverage counter* counts how many miscoverage events happened in a row until time $t$:

$$\text{MC}_t = \begin{cases} \text{MC}_{t-1} + 1, & Y_t \notin \hat{C}_t(X_t) \\ 0, & \text{otherwise} \end{cases},$$

where $\text{MC}_0 = 0$. Similarly to the coverage metric, $\text{MC}_t = 0$ at timestamps for which $Y_t \in \hat{C}_t(X_t)$. By contrast, when $Y_t \notin \hat{C}_t(X_t)$, the value of $\text{MC}_t$ is the length of the sequence of previous miscoverage events. Therefore, we can apply `Rolling RC` to control the `MC` level and prevent long sequences of failures. Interestingly, controlling the miscoverage counter immediately grants a control over the standard coverage metric, as stated next.

**Proposition 1.** *If the `MC` risk is at most $\alpha$, then the miscoverage risk is at most $\alpha$.*

In Appendix A.5 we provide the proof of this proposition and in Section E.2 we explain how to choose the nominal `MC` level to achieve a given coverage rate $1 - \alpha$. In a nutshell, we argue that a model that has access to the true conditional quantiles attains an `MC` of $\alpha/(1 - \alpha)$. Therefore, in our experiments we seek to form the tightest intervals with an `MC` risk controlled at this level.

**MSL:** As implied by its name, the metric *miscoverage streak length* evaluates the average length of miscoverage streaks of the constructed prediction intervals. In contrast to `MC`, which is defined on a

single timestamp, the MSL is defined over a sequence of uncertainty sets $\{\hat{C}_t(X_t)\}_{t=T_0}^{T_1} \subseteq 2^{\mathcal{Y}}$ and response variables $\{Y_t\}_{t=T_0}^{T_1} \subseteq \mathcal{Y}$ as:

$$\text{MSL} := \frac{1}{|\mathcal{I}|} \sum_{t \in \mathcal{I}} \min\{i : Y_{t+i} \in \hat{C}_{t+i}(X_{t+i}) \text{ or } t = T_1\},$$

where $\mathcal{I}$ is a set containing the starting times of all miscoverage streaks. The formal description is given in Appendix E.1, where we also show that an ideal model that has access to the true conditional quantiles attains an MSL of $1/(1-\alpha)$. Therefore, we seek to produce the narrowest intervals having an MSL close to this value.

### 4.1.2 CONTROLLING THE BINARY LOSS

In this section, we focus on the more standard long-range coverage loss as in ACI (Gibbs & Candes, 2021). We test the performance of Rolling RC on five real-world benchmark data sets with a 1-dimensional $Y$: Power, Energy, Traffic, Wind, and Prices. We commence by fitting an initial quantile regression model on the first 5000 data points, to obtain a reasonable predictive system. Then, passing time step 5001, we start applying the calibration procedure while continuing to fit the model in an online fashion; we keep doing so until reaching time step 20000. Lastly, we measure the performance of the deployed calibration method on data points corresponding to time steps 8001 to 20000. In all experiments, we fit an LSTM predictive model (Hochreiter & Schmidhuber, 1997) in an online fashion, minimizing the pinball loss to estimate the 0.05 and 0.95 conditional quantiles of $Y_t \mid X_t$; these estimates are used to construct prediction intervals with target 90% coverage rate. We calibrate the intervals according to (6) and examine two options for the stretching function: (i) no stretching, and (ii) 'error adaptive' stretching, described in Section 3.2.1. Appendix C.1.1 provides more details regarding the data sets and this experimental setup.

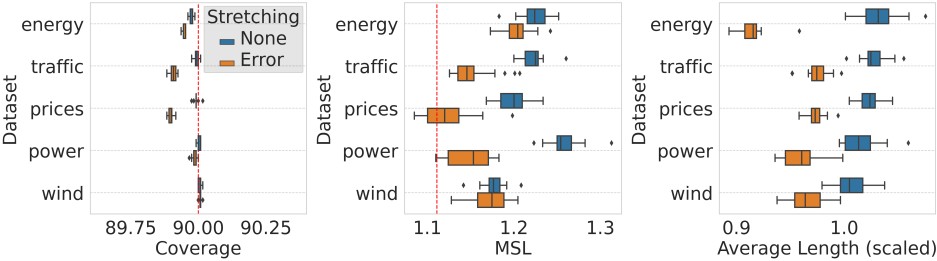

Figure 3: Performance of Rolling RC on real data sets, aiming to control the coverage rate at level $1 - \alpha = 90\%$. The length of the prediction intervals is scaled per data set by the average length of the constructed intervals. Results are evaluated on 20 random initializations of the predictive model.

Figure 3 summarizes the performance metrics presented in Section 4.1.1, showing that both stretching methods attain the desired coverage level; this is guaranteed by Theorem 1. Additionally, this figure indicates that Rolling RC with the 'error adaptive' stretching constructs narrower intervals with better conditional coverage compared to Rolling RC applied without stretching, as indicated by the MSL metric. In Appendix D.1 we compare Rolling RC to a calibration-set-based method and evaluate the methods using another metric for evaluating conditional coverage for time-series data. These experiments show that constructing uncertainty sets using the conformity scores of a calibration set performs worse than Rolling RC with or without stretching.

### 4.1.3 CONTROLLING MISCOVERAGE COUNTER

Figure 4c shows that Rolling RC applied on the real data sets with the goal of controlling the long-range coverage at level $1 - \alpha = 90\%$ achieves MC risk that is higher than $\alpha/(1-\alpha) = 1/9$. Following the discussion in Section 4.1.1, this indicates that the constructed intervals tend to miscover one or more response variables in consecutive data points. To alleviate this, we repeat the same experiment in Section 4.1.2, but apply Rolling RC to control the MC at level $r = 1/9$. The results are summarized in Figure 4, revealing that (i) the MC risk is rigorously controlled even though it is defined over the time horizon, and (ii) by controlling the MC risk we also achieve valid coverage rate.

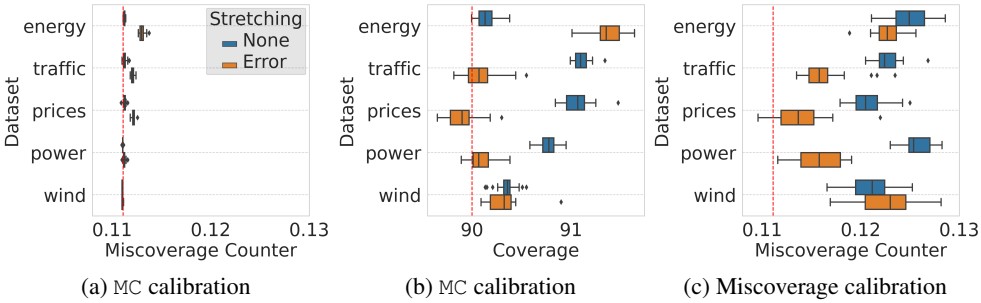

| (a) `MC` calibration | (b) `MC` calibration | (c) Miscoverage calibration |

Figure 4: Performance of `Rolling RC` on real data sets. In (a) and (b) we aim to control the `MC` risk at level $r = \alpha/(1-\alpha) = 1/9$. In (c) we aim to control the coverage at level $1 - \alpha = 90\%$. Other details are as in Figure 3.

## 4.2 HIGH DIMENSIONAL RESPONSE: CONTROLLING MULTIPLE RISKS

In this section, we analyze `Rolling RC` for multiple risks in the depth prediction setting. In particular, we follow the protocol described in Section 1.1 and apply the multiple risks controlling method from Section 3.3 to control the image miscoverage rate defined in (2) at level $20\%$ and the center failure rate from (7) at level $10\%$. We construct the intervals according to (9) from Appendix B, using exponential stretching where the vector $\underline{\theta}_t$ is aggregated into a scalar by taking the maximal coordinate in this vector. We repeat this experiment for 10 trials. In Appendix C.2 we provide the full details about this experimental setup.

Figure 5 displays the results of `Rolling RC` obtained by controlling (i) only the image miscoverage loss, and (ii) both the image miscoverage loss and the center failure loss. As portrayed, when `Rolling RC` is set to control only the image miscoverage risk, it violates the center failure loss; the center coverage falls below $60\%$ for $17\%$ of the time-steps. However, when applying `Rolling RC` to control the two risks, it achieves both a valid image coverage rate, of approximately $85.6\%$, and a valid center failure rate, of $9.9\%$. This is not a surprise, as it is guaranteed by Theorem 2.

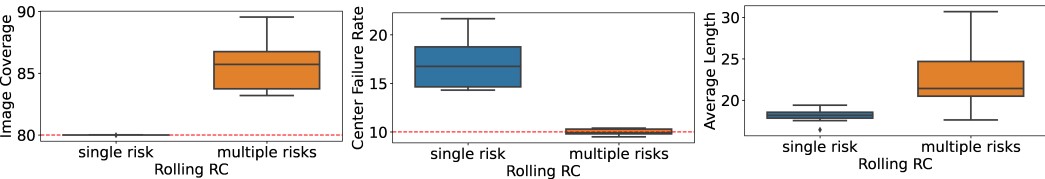

Figure 5: Performance of `Rolling RC` applied to control only the 'image coverage' (single risk) or both 'image coverage' and 'center failure' (multiple risks).

## 5 CONCLUSION

In this paper, we introduced `Rolling RC`, a novel method for quantifying prediction uncertainty for any time-series data using any online learning model. Our proposal is guaranteed to achieve any desired level of risk (1) without making assumptions on the data, and can be applied for a broad class of tasks, such as regression, classification, image-to-image regression, and more. Furthermore, in Section 3.3 we extended `Rolling RC` to provably control multiple risks, so that the uncertainty sets it constructs are valid for all given risks. One limitation of our method is the reliance on a fixed step size $\gamma$, used to tune the raw risk level; improper choice of this parameter may introduce undesired delays in adapting to distributional shifts. Therefore, it is of great interest to develop a procedure that would automatically choose $\gamma$, e.g., by borrowing ideas from (Gibbs & Candès, 2022; Zaffran et al., 2022). Meanwhile, we suggested a way to overcome this limitation using a stretching function $\varphi$, which leads to improved performance, as indicated by the experiments.

ETHICS STATEMENT

While valid calibration is an important step toward making learning systems safer, fairer, and more robust, we emphasize that calibration is not a panacea and one must always treat data with care—especially when data-driven predictions are used for high-stakes decisions. In particular, the flexibility of our proposed method is a mixed blessing, in that the system builder has the ability and responsibility to make important design choices. These choices can have important consequences, so they must be treated with care. We offer initial guidelines herein, acknowledging that of course, they do not anticipate the consequences in all possible use cases. Nonetheless, we highlight that uncertainty quantification is an important component to designing learning systems to have a positive impact in the real world, where the data are complex and distributions are continually shifting. Our work makes one step toward this goal.

For reproducibility, our code carrying out the experiments herein is included as Supplementary Material with this submission.

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

# Appendices

## A THEORETICAL RESULTS

### A.1 PROOF OF THEOREM 1

The proof of Theorem 1 is based on that of Proposition 4.1 in (Gibbs & Candes, 2021). While the proof is similar, our work greatly enlarges the scope of the result.

We begin by showing that $\theta_t$ is bounded throughout the entire calibration process. For this purpose, we assume that the loss satisfies that for all $y \in \mathcal{Y}$ and $C \in 2^{\mathcal{Y}}$: $L(y, C) \in [-B, B]$ where $B > 0$ is a real number.

**Lemma 1.** *Under the assumptions of Theorem 1, for all $t \in \mathbb{N}$, $\theta_t \in [m - \gamma 2B, M + \gamma 2B]$.*

*Proof.* Assume for the sake of contradiction that there exists $t \in \mathbb{N}$ such that $\theta_t > M + 2\gamma B$ (the complementary case is similar). Further assume that for all $t' < t$: $\theta_{t'} \leq M + 2\gamma B$. Since $l_t, r \in [-B, B]$, we get that:

$$\theta_{t-1} = \theta_t - \gamma(l_t - r) \geq \theta_t - 2\gamma B > M + 2\gamma B - 2\gamma B = M.$$

Therefore, $\theta_{t-1} > M$. Since $L(y, f(X, \theta, \mathcal{M}) = \mathcal{Y}) < r$ for $\theta > M$, we get that $l_t < r$. As a result:

$$\theta_t = \theta_{t-1} + \gamma(l_t - r) < \theta_{t-1} \leq M + \gamma 2B.$$

Which is a contradiction to our assumption. □

Next, we prove Theorem 1.

*Proof.* By applying Lemma 1 we get that $\theta_t \in [m - \gamma 2B, M + \gamma 2B]$ for all $t \in \mathbb{N}$. Denote $m' = m - \gamma 2B$, and $M' = M + \gamma 2B$. We follow the proof of (Gibbs & Candes, 2021, Proposition 4.1) and expand the recursion defined in (5):

$$[m', M'] \ni \theta_{T+1} = \theta_1 + \sum_{t=1}^{T} \gamma(l_t - r).$$

By rearranging this we get that:

$$\frac{m' - \theta_1}{T\gamma} \leq \frac{1}{T}\sum_{t=1}^{T}(l_t - r) = \frac{\theta_{T+1} - \theta_1}{T\gamma} \leq \frac{M' - \theta_1}{T\gamma}.$$

Therefore:

$$\left| \frac{1}{T}\sum_{t=1}^{T}(l_t - r) \right| \leq \frac{\max\{\theta_1 - m', M' - \theta_1\}}{T\gamma}.$$

Lastly, the definition of the loss, $l_t = L(Y_t, \hat{C}_t(X_t))$, gives us the risk statement in (1). □

Notice that the above proof additionally implies a finite-sample bound for the deviation of the empirical risk from the desired level. In particular, the average loss is within a $C/T$ factor of $r$, where $C = (M' - m')/\gamma = (M - m + 4 \cdot \gamma B)/\gamma$. This bound is deterministic, not probabilistic. Thus, even for the most erratic input sequences, the method has an average loss very close to the nominal level.

### A.2 GENERAL VERSION OF THEOREM 1

In this section, we provide a general statement with more abstract notations of Theorem 1. The following notations extend our proposal to a broader class of problems, so that it could be applied for a larger set of tasks, such as multi-label classification. Here, we assume that the function $f$ generates variables in $\mathcal{Y}'$ and that the loss function is defined as $L : (\mathcal{Y}, \mathcal{Y}') \to \mathbb{R}$. Furthermore, we assume that there exist a minimal value $\underline{\mathcal{y}}' \in \mathcal{Y}'$ and a maximal value $\bar{\mathcal{y}}' \in \mathcal{Y}'$ for which $L(y, \underline{\mathcal{y}}') > r$ and $L(y, \bar{\mathcal{y}}') < r$. In the main text, we set $\mathcal{Y}' = 2^{\mathcal{Y}}$, $\underline{\mathcal{y}}' = \emptyset$ and $\bar{\mathcal{y}}' = \mathcal{Y}$. We now show that the risk is controlled at the desired level even under this general setting.

**Theorem 4.** *Suppose that $f : (\mathcal{X}, \mathbb{R}, \mathbb{M}) \to \mathcal{Y}'$ is an interval/set constructing function. In addition, suppose that there exist constants $m$ and $M$ such that for all $x \in \mathcal{X}$, $y \in \mathcal{Y}$ and $\mathcal{M} \in \mathbb{M}$, $f(x, \theta, \mathcal{M}) = \underline{\mathcal{Y}}'$ for all $\theta > M$, $f(X, \theta, \mathcal{M}) = \bar{\mathcal{Y}}'$ for all $\theta < m$. Further suppose that the loss is bounded and satisfies $L(y, \bar{\mathcal{Y}}') < r$ and $L(y, \underline{\mathcal{Y}}') > r$. Consider the following series of calibrated intervals: $\{\hat{C}_t(X_t)\}_{t=1}^{\infty}$, where $\hat{C}_t(X_t)$ is defined according to (4). Then, the calibrated intervals satisfy the risk requirement in (1).*

*Proof.* The proof is similar to the one of Theorem 1 and hence omitted. $\square$

## A.3 PROOF OF THEOREM 2

The proof of Theorem 2 is similar to the proof of Theorem 1. We assume that all losses $\{L_i\}_{i=1}^{k}$ are bounded in the interval $[-B, B]$, as in Section A.1, and begin by showing that all coordinates in $\underline{\theta}_t$ are upper bounded.

**Lemma 2.** *Under the assumptions of Theorem 2, for all $t \in \mathbb{N}$ and $i \in \{1, ..., k\}$, $\underline{\theta}_t^i \leq M + \gamma 2B$.*

*Proof.* Assume for the sake of contradiction that there exist $t \in \mathbb{N}$ and $i \in \{1, ..., k\}$ such that $\underline{\theta}_t^i > M + 2\gamma B$. Further assume that for all $t' < t$: $\underline{\theta}_{t'}^i \leq M + 2\gamma B$. Since $l_t^i, r^i \in [-B, B]$, we get that:

$$\underline{\theta}_{t-1}^i = \underline{\theta}_t^i - \gamma(l_t^i - r^i) \geq \underline{\theta}_t^i - 2\gamma B > M + 2\gamma B - 2\gamma B = M.$$

Therefore, $\underline{\theta}_{t-1}^i > M$. Since $L^i(y, f(X, \underline{\theta}, \mathcal{M}) = \underline{\mathcal{Y}}) < r^i$ for $\underline{\theta}^i > M$, we get that $l_t^i < r^i$. As a result:

$$\underline{\theta}_t^i = \underline{\theta}_{t-1}^i + \gamma(l_t^i - r) < \underline{\theta}_{t-1}^i \leq M + \gamma 2B.$$

Which is a contradiction to our assumption. $\square$

Next, we prove Theorem 2.

*Proof.* By applying Lemma 2 we get that $\underline{\theta}_t^i \leq M + \gamma 2B$ for all $t \in \mathbb{N}$ and $i \in \{1, ..., k\}$. Denote $M' = M + \gamma 2B$. We expand the recursion defined in (5):

$$\underline{\theta}_{T+1}^i = \underline{\theta}_1^i + \sum_{t=1}^{T} \gamma(l_t^i - r^i) \leq M'.$$

By rearranging this we get that:

$$\frac{1}{T} \sum_{t=1}^{T} l_t^i - r^i = \frac{1}{T} \sum_{t=1}^{T} (l_t^i - r^i) = \frac{\underline{\theta}_{T+1}^i - \underline{\theta}_1^i}{T\gamma} \leq \frac{M' - \underline{\theta}_1^i}{T\gamma}.$$

Therefore:

$$\frac{1}{T} \sum_{t=1}^{T} l_t^i \leq r^i + \frac{M' - \underline{\theta}_1^i}{T\gamma}.$$

Lastly, by the definition of the loss, $l_t^i = L^i(Y_t, \hat{C}_t(X_t))$ and by setting $D^i = \frac{M' - \underline{\theta}_1^i}{\gamma}$, we get the statement in Theorem 2. $\square$

## A.4 PROOF OF THEOREM 3

The proof of Theorem 3 is similar to the proof of Theorem 2. We assume that all losses $\{L_i\}_{i=1}^{k}$ are bounded in the interval $[-B, B]$, as in Section A.1, and begin by showing that all coordinates in $\underline{\theta}_t$ are lower bounded.

**Lemma 3.** *Under the assumptions of Theorem 3, for all $t \in \mathbb{N}$ and $i \in \{1, ..., k\}$, $\underline{\theta}_t^i \geq m - \gamma 2B$.*

*Proof.* Assume for the sake of contradiction that there exist $t \in \mathbb{N}$ and $i \in \{1, ..., k\}$ such that $\underline{\theta}_t^i < m - 2\gamma B$. Further assume that for all $t' < t$: $\underline{\theta}_{t'}^i \geq m - 2\gamma B$. Since $l_t^i, r^i \in [-B, B]$, we get that:

$$\underline{\theta}_{t-1}^i = \underline{\theta}_t^i - \gamma(l_t^i - r^i) \leq \underline{\theta}_t^i + 2\gamma B < m - 2\gamma B + 2\gamma B = m.$$

Therefore, $\underline{\theta}_{t-1}^i < m$. Since $L^i(y, f(X, \underline{\theta}, \mathcal{M}) = \mathcal{Y}) > r^i$ for $\underline{\theta}^i < m$, we get that $l_t^i > r^i$. As a result:

$$\underline{\theta}_t^i = \underline{\theta}_{t-1}^i + \gamma(l_t^i - r) > \underline{\theta}_{t-1}^i > m - \gamma 2B.$$

Which is a contradiction to our assumption. $\qquad\square$

Next, we prove Theorem 3.

*Proof.* By applying Lemma 2 and Lemma 3 we get that $\underline{\theta}_t^i \in [m - \gamma 2B, M + \gamma 2B]$ for all $t \in \mathbb{N}$ and $i \in \{1, ..., k\}$. Denote $m' = m - \gamma 2B$ and $M' = M + \gamma 2B$. We expand the recursion defined in (5):

$$[m', M'] \ni \underline{\theta}_{T+1}^i = \underline{\theta}_1^i + \sum_{t=1}^T \gamma(l_t^i - r^i).$$

By rearranging this we get that:

$$\frac{m' - \underline{\theta}_1^i}{T\gamma} \leq \frac{1}{T}\sum_{t=1}^T (l_t^i - r^i) = \frac{\underline{\theta}_{T+1}^i - \underline{\theta}_1^i}{T\gamma} \leq \frac{M' - \underline{\theta}_1^i}{T\gamma}.$$

Therefore:

$$\left| \frac{1}{T}\sum_{t=1}^T (l_t^i - r^i) \right| \leq \frac{\max\{\underline{\theta}_1^i - m', M' - \underline{\theta}_1^i\}}{T\gamma}.$$

Lastly, the definition of the loss, $l_t^i = L^i(Y_t, \hat{C}_t(X_t))$, gives us the statement in Theorem 3. $\qquad\square$

### A.5 Proof of Proposition 1

*Proof.* $\mathbb{1}\{Y_t \notin \hat{C}_t(X_t)\} \leq MC_t$ for any $t \in \mathbb{N}$. Therefore:

$$\lim_{T\to\infty} \frac{1}{T}\sum_{t=1}^T MC_t \leq \alpha \implies \lim_{T\to\infty} \frac{1}{T}\sum_{t=1}^T \mathbb{1}\{Y_t \notin \hat{C}_t(X_t)\} \leq \alpha$$

$\qquad\square$

## B Uncertainty Quantification in Online Image-to-Image Regression Problems

### B.1 General Formulation

Recall the depth estimation problem from Section 1.1, where we construct per-pixel prediction intervals by online processing an incoming video stream. In what follows, we discuss such image-to-image regression problems more generally, providing a scheme to construct and calibrate pixel-valued intervals. Consider a running model $\mathcal{M}_t(X_t)$ that maps the input image $X_t$ to a point prediction of $Y_t$, and we wish to control the image miscoverage loss $L_{\text{image miscoverage}}$ from (2). We take inspiration from Angelopoulos et al. (2022b) and form the prediction intervals around each pixel $(m, n)$ of the estimated image $\mathcal{M}_t(X_t)$ as

$$\hat{C}_t^{m,n}(X_t) = f^{m,n}(X_t, \theta_t, \mathcal{M}_t) = [\mathcal{M}_t^{m,n}(X_t) - \lambda_t l_t^{m,n}(X_t), \mathcal{M}_t^{m,n}(X_t) + \lambda_t u_t^{m,n}(X_t)]. \quad (9)$$

Above, $l_t^{m,n}(X_t)$ and $u_t^{m,n}(X_t)$ represent the uncertainty in the lower and upper directions, respectively. That is, a large value of $l_t^{m,n}(X_t)$ indicates that the pixel has a high uncertainty in the upper direction. Similarly, a large value of $u_t^{m,n}(X_t)$ indicates that the pixel has a high uncertainty in the lower direction. A natural choice for the lower and upper uncertainty measures is a model estimating the absolute residual error per-pixel, given by $u_t^{m,n}(X_t) = l_t^{m,n}(X_t) = |Y_t^{m,n} - \mathcal{M}_t^{m,n}(X_t)|$. We

provide more sophisticated examples for such uncertainty functions in Section B.2. The parameter $\lambda_t = \varphi(\theta_t) \in \mathbb{R}$ in (9) stretches the calibration parameter $\theta_t$, which we update according to (5). Importantly, this procedure is an instantiation of `Rolling RC` and thus attains the correct image coverage, as guaranteed by Theorem 1. This is also validated in the experiment from Section 1.1.

### B.2  UNCERTAINTY QUANTIFICATION HEURISTICS

In this section, we present possible choices for the uncertainty heuristics for the interval constructing function given in (9).

#### B.2.1  BASELINE CONSTANT

The most naive choice for an uncertainty heuristic is outputting a constant value for every $m, n, X$: $l^{m,n}(X) = u^{m,n}(X) = 1$. In other words, the set constructing function is defined as:

$$f^{m,n}(X_t, \theta_t, \mathcal{M}_t) = [\mathcal{M}_t^{m,n}(X_t) - \lambda_t, \mathcal{M}_t^{m,n}(X_t) + \lambda_t].$$

This approach has two main limitations: (i) the calibrated intervals are symmetric, having the same uncertainty size in both the upper and lower directions, and (ii) all the pixel-valued intervals have the same length. These limitations lead to unnecessarily wide intervals that are less informative. We show how to overcome these limitations with the methods presented hereafter.

#### B.2.2  MAGNITUDE OF THE RESIDUAL

The residual magnitude heuristic was introduced by Angelopoulos et al. (2022b) as a simple uncertainty quantification technique. Here, $l^{m,n}(X) = u^{m,n}(X) = \hat{r}^{m,n}(x)$ is an estimate for the residual $|\mathcal{M}^{m,n}(X) - Y^{m,n}|$ and it is formulated as an online learning model fitted to minimize the squared residual loss, given by $(\hat{r}(x) - |\mathcal{M}^{m,n}(x) - y|)^2$. An ideal model that minimizes this loss function outputs the exact residual: $\hat{r}(x) = |\mathcal{M}^{m,n}(x) - y|$ and thus achieves 100% coverage rate for $\lambda = 1$. In practice, however, the fitted model $\hat{r}$ may not be accurate and thus we apply `Rolling RC`, to ensure valid risk control. Observe that, unlike the constant heuristic, here, each pixel is assigned a different uncertainty size. Nevertheless, both techniques produce symmetric prediction intervals.

#### B.2.3  PREVIOUS RESIDUALS

In contrast to the residual's magnitude heuristic, here, we take advantage of the online setting in which the data set is received as a stream, and use the past residuals to estimate the current one. We define the positive and negative residuals at time $t$ as:

$$r_t^{m,n} = \mathcal{M}^{m,n}(X_t) - Y_t^{m,n},$$
$$r_t^{m,n+} = \max\{r_t^{m,n}, 0\},$$
$$r_t^{m,n-} = \max\{-r_t^{m,n}, 0\}.$$

The uncertainty heuristic in the lower (upper) direction is formulated as the average of the positive (negative) residual in the previous $p$ time-steps:

$$l_t^{m,n}(X_t) = \frac{1}{p}\sum_{t'=t-p}^{t-1} r_t^{m,n+},$$

$$u_t^{m,n}(X_t) = \frac{1}{p}\sum_{t'=t-p}^{t-1} r_t^{m,n-}. \tag{10}$$

In our experiments, we set the sliding window's size to $p = 5$.

#### B.2.4  CORRECTING PIXELS DISPLACEMENTS WITH IMAGE REGISTRATION

The 'previous residuals' method suffers from the following crucial limitation. Objects in the response image $Y$ may appear in different positions across time, so that an object that lies in pixel $(m, n)$ at frame $t$ might appear in a different pixel at time $t + 1$, e.g., $(m + 7, n - 11)$. For example, in our

depth prediction example the camera and the depth sensor move during the online process, so objects change their locations and do not remain in a fixed pixel between consecutive frames. Therefore, to obtain more accurate residual estimates it is better to correct for the displacement of pixels in consecutive frames. For this purpose, we apply an image registration algorithm before evaluating the residuals. In particular, we use optical flow ($\texttt{OF}$) to register the estimated depth images and the ground truth depth maps. Suppose that $\texttt{OF}(\text{im}_1, \text{im}_2)$ receives two images as an input and returns the result of the registration of the first image $\text{im}_1$ to the second one $\text{im}_2$. We recursively define optical flow on a sequence as:

$$\texttt{OF}^{\text{seq}}(\text{im}_1, \emptyset) = \text{im}_1,$$

$$\texttt{OF}^{\text{seq}}(\text{im}_1, \{\text{im}_i\}_{i=2}^{k}) = \texttt{OF}^{\text{seq}}(\texttt{OF}(\text{im}_1, \text{im}_2), \{\text{im}_i\}_{i=3}^{k}).$$

Then, we register the previous estimated depth images and ground truth depth maps

$$\mathcal{M}^{\text{reg}}(X_{t-i}) = \texttt{OF}^{\text{seq}}(\mathcal{M}(X_{t-i}), \{\mathcal{M}(X_{t-i+j})\}_{j=1}^{i-1}),$$

$$Y_{t-i}^{\text{reg}} = \texttt{OF}^{\text{seq}}(Y_{t-i}, \{Y_{t-i}\}_{j=1}^{i-1}).$$

In plain words, we register each image using the next ones in the sequence. We define the registered residuals as:

$$\bar{r}_t = \mathcal{M}^{\text{reg}}(X_t) - Y_t^{\text{reg}},$$

$$\bar{r}_t^{m,n+} = \max\{\bar{r}_t^{m,n}, 0\},$$

$$\bar{r}_t^{m,n-} = \max\{-\bar{r}_t^{m,n}, 0\}.$$

Then, we compute the average residual, as in (10):

$$l_t^{m,n}(X_t) = \frac{1}{p} \sum_{t'=t-p}^{t-1} \bar{r}_t^{m,n+},$$

$$u_t^{m,n}(X_t) = \frac{1}{p} \sum_{t'=t-p}^{t-1} \bar{r}_t^{m,n-}.$$

This displacement consideration indeed improves the performance, as indicated by the experiments in Section D.2.

## C EXPERIMENTAL SETUP

### C.1 SINGLE-OUTPUT TASKS

#### C.1.1 THE QUANTILE REGRESSION MODEL'S ARCHITECTURE

The neural network architecture is composed of four parts: an MLP, an LSTM, and another two MLPs. To estimate the uncertainty of $Y_t \mid X_t$, we first map the previous $k$ samples $\{(X_{t-i}, Y_{t-i}, \tau)\}_{i=1}^{k}$ through the first MLP, denoted as $f_1$,

$$w_{t-i}^1 = f_1(x_{t-i}, y_{t-i}, \tau),$$

where we set $k$ to 3 in our experiments. The outputs are then forwarded through the LSTM network, denoted as $f_2$:

$$\{w_{t-i}^2\}_{i=1}^{k} = f_2(\{w_{t-i}^1\}_{i=1}^{k}).$$

Note that since $f_2$ is an LSTM model, $w_{t-i}^2$ is used to compute $w_{t-i+1}^2$. The last output $w_{t-1}^2$ of the LSTM model, being an aggregation of the previous $k$ samples, is fed, together with $(X_t, \tau)$, to the second MLP model, denoted as $f_3$:

$$w_t^3 = f_3(w_{t-1}^2, X_t).$$

Lastly, we pass $w_t^3$ through the third MLP, denoted by $f_4$, with one hidden layer that contains 32 neurons:

$$\hat{q}_\tau(X_t) = f_4(w_t^3, \tau).$$

The networks contain dropout layers with a parameter equal to 0.1. The model's optimizer is Adam (Kingma & Ba, 2015) and the batch size is 512, i.e., the model is fitted on the most recent 512 samples in each time step. Before forwarding the input to the model, the feature vectors and response variables were normalized to have unit variance and zero mean using the first 8000 samples of the data stream.

### C.1.2 TRAINING THE QUANTILE REGRESSION MODEL

Estimating the conditional quantile function can be done, for example, by minimizing the pinball loss in lieu of the standard mean squared error loss used in classic regression; see (Izbicki et al., 2020; Jia & Jeong, 2022; Koenker & Bassett, 1978; Koenker & Hallock, 2001; Meinshausen, 2006). Specifically, in our experiments with time-series data we minimize the objective function:

$$\min_{\mathcal{M}_t} \sum_{t'=1}^{t} \rho_{\alpha/2}(Y_{t'}, \mathcal{M}_t(X_{t'}, \alpha/2)) + \rho_{1-\alpha/2}(Y_{t'}, \mathcal{M}_t(X_{t'}, \alpha/2)),$$

where

$$\rho_\alpha(y, \hat{y}) = \begin{cases} \alpha(y - \hat{y}) & y - \hat{y} > 0, \\ (1 - \alpha)(\hat{y} - y) & \text{otherwise.} \end{cases}$$

is the pinball loss. Since the data points arrive sequentially, we formulate $\mathcal{M}_t$ as an LSTM model and minimize the above cost function in an online fashion as follows. Given a new labeled test point $(X_t, Y_t)$, we (i) compute the pinball loss both for the lower and upper quantiles, i.e., $\rho_{\alpha/2}(Y_t, \mathcal{M}_t(X_t, \alpha/2))$ and $\rho_{1-\alpha/2}(Y_t, \mathcal{M}_t(X_t, 1 - \alpha/2))$, respectively; and (ii) update the parameters of the LSTM model $\mathcal{M}_t$ by applying a few gradient steps with ADAM optimizer. More details on the network architecture are given in Appendix C.1.1.

### C.1.3 HYPER-PARAMETERS TUNING

For both real and synthetic data sets, we examined all combinations of the raw model's hyperparameters (with no calibration applied) on one initialization of the model, and chose the setting in which the model attained the smallest pinball loss, evaluated on the validation set, indexed by $6001 - 8000$. The combinations we tested are presented in Table 1. Some of the configurations required more than 11GB of memory to train the model, so we did not consider them in our experiments. The chosen configuration was later used for choosing the calibration's learning rate $\gamma$, as explained next. We note that using the validation set to tune the hyperparameters and the choice of the stretching function is a heuristic. Yet, we found this rule of thumb to work well empirically, as visualized in Figure 7 in Appendix D.1.1. Of course, there are situations where this approach would not be most effective since the data is not i.i.d., but we believe it is a sensible suggestion. Another advantage of tuning the hyperparameters this way is that it facilitates the comparison between the different methods since they all follow the same automatic tuning approach.

The updating rates we tested are: $\gamma \in \{0.025, 0.03, 0.05, 0.09, 0.1, 0.15, 0.2, 0.35\}$. We chose $\gamma$ that attained the smallest pinball loss, evaluated on points $5001 - 8000$.

In Appendix C.4 we explain how we chose the hyper-parameters for the stretching functions.

### C.2 THE DEPTH PREDICTION SETUP

### C.2.1 DATA SET AND AUGMENTATIONS

We used the KITTI data set which contains pairs of a colored (RGB) image (Geiger et al., 2013) and a latent ground truth depth map (Uhrig et al., 2017). We filled the missing depth values with the colorization algorithm developed by Levin et al. (2004). Then, we scaled the depth values to the range [0,10]. We augmented the images according to the following protocol. Images and depths used for training the model were resized using one of the following ratios [0.5, 0.6, 0.7, 0.8, 0.9, 1.0, 1.1, 1.2, 1.3, 1.4, 1.5], chosen with equal probability, cropped randomly and re-scaled to $448 \times 448$, and then flipped horizontally with a 50% chance. Images and depths used for testing and updating the calibration model were resized with a ratio of 0.5 and re-scaled to $448 \times 448$. In both cases, we augmented the colors of the RGB image and blurred it, as described in (Yin et al., 2021). Notice that the augmentations are deterministic during inference and random during training. Therefore, the augmentations are chosen randomly in each trial. The main purpose of these augmentations is to improve the model's training.

### C.2.2 THE DEPTH PREDICTION MODEL

The base prediction model $\mathcal{M}$ we used is LeReS (Yin et al., 2021) with ResNeXt101 backbone initialized with the pre-trained network from `https://cloudstor.aarnet.edu.au/plus/`

`s/lTIJF4vrvHCAI31`. This pre-trained network was not fitted on the KITTI data set used in our experiments. We fitted the model offline on the first 6000 samples for 60 epochs, to obtain a reasonable predictive system. Then, passing time step 6001 we start training the model in an online fashion while applying the calibration procedure. Lastly, we measure the performance of the deployed calibration method on data points corresponding to time steps 8001 to 10000.

### C.2.3 Training The Uncertainty Quantification Models

For each uncertainty quantification model presented in Section B.2, we used a pre-trained LeReS (Yin et al., 2021) network as the base network architecture and initialized the last two components of the network with random weights. We simultaneously trained the uncertainty quantification model and the LeReS depth model on the first 6000 samples for 60 epochs, and in an online fashion at timestamps 6001 to 10000, as mentioned in Section C.2.2.

### C.2.4 Depth Example Setup

In this section, we describe the setup we used for the depth experiment presented in Section 1.1 in the main text. As explained in Section C.2.2, we used LeReS (Yin et al., 2021) as the base depth prediction model. The model's predictions were calibrated by our `Rolling RC`, with the design choice for $f$ given in Section 9. The uncertainty heuristics we used in this experiment are the previous residuals with registration, as described in Section B.2.4, with a sliding window of size $p = 5$. We used scikit-image's implementation of optical flow. We set the parameter "num_iter" to 2 and the "num_warp" to 1, to reduce the computational complexity. The figures are taken from one trial of the experiment, for which the random seed value was set to 0. The only hyper-parameter we tuned in this experiment is the calibration's learning rate $\gamma$ which we tuned in the following way. We applied `Rolling RC` with the following learning rates: $\gamma \in \{0.001, 0.005, 0.01, 0.05, 0.1, 0.2, 0.5, 1, 2, 10\}$ and chose the $\gamma$ that achieved the narrowest intervals among those with coverage greater than 79.9% on the validation set. The validation samples are those that correspond to timestamps 7001 to 8000. The value that was finally chosen is $\gamma = 0.2$.

### C.2.5 Multiple Risks Controlling Setup

In this section, we describe the setup we used for the multiple risks controlling experiment presented in Section 4.2. We followed the experimental protocol described in Section C.2.4 except for the tuning of $\gamma$. When `Rolling RC` is applied to control multiple risks, the learning rate $\gamma$ is a vector, so we examine all possible choices of $\gamma^1, \gamma^2 \in \{0.00001, 0.0001, 0.001, 0.005, 0.01, 0.1, 0.5, 2\}$. We chose the vector $\gamma$ that achieved the narrowest intervals among those with coverage greater than 79.9% and center failure rate lower than 11%, evaluated on the validation samples, corresponding to timestamps 7001 to 8000. In this experiment, we set $\lambda_t$ in (9) to be the maximal value in the vector $\underline{\theta}_t$. That is, we used the 'max aggregation' described in Section D.3.

### C.2.6 Implementation Details

In this section, we describe the technical details of implementing the depth prediction model and the uncertainty quantification heuristics. Popular depth prediction models estimate the depth up to an unknown scale and shift (Li et al., 2022a;b; Yuan et al., 2022). That is, an ideal model $\mathcal{M}$ satisfies that for every $X_t \in \mathcal{X}$ there exist $\mu_X, \sigma_X \in \mathbb{R}$ such that:

$$\sigma_t \mathcal{M}(X_t) + \mu_t = Y_t.$$

We correct the model's outputs to actual depth estimates in the following way. First, we assume that we are given the ground truth depth of a small set of pixels. Then, we use this information to estimate the scale and shift: when $X_t$ is revealed, we uniformly choose 200 pixels of it and assume that their depth is given along with $X_t$. Next, we obtain the learning model's output $\mathcal{M}(X_t)$ and apply least squares polynomial fitting to compute the estimated scale $\hat{\mu}_t$ and shift $\hat{\sigma}_t$. Finally, we produce the following depth estimate:

$$\hat{Y}_t = \hat{\sigma}_t \mathcal{M}_t(X_t) + \hat{\mu}_t. \tag{11}$$

Throughout this paper, we consider the quantity in (11) as the output of the depth model $\mathcal{M}$.

We utilize the sparse ground truth depth map to correct the estimates of the uncertainty heuristics as well. Recall that the residual heuristic from Section B.2.2 produces an estimate $\hat{r}(X_t)$ for the residual

$|\mathcal{M}^{m,n}(X_t) - Y_t|$, where $\mathcal{M}(X_t)$ is the scaled model's prediction. We compute the scale and shift for the residual's prediction via polynomial fitting, and output the scaled residual, as in (11). Similarly, we correct the previous residuals heuristic defined in Section B.2.3 by re-scaling the positive and negative residuals.

Another important technical detail is dealing with invalid pixels. Invalid pixels are pixels with depth that is too small (below $10^{-8}$), or pixels that are padded to the image. We do not consider these pixels for updating the calibration scheme or for evaluating the methods' performance. For instance, the image coverage rate is practically the coverage rate over all valid pixels in a given image.

Lastly, throughout the depth prediction experiments we used the following formulation of the exponential stretching function for `Rolling RC`:

$$\varphi^{\text{exp.}}(x) = \begin{cases} e^x - 1, & x > 0.1, \\ x, & -0.1 \le x \le 0.1, \\ -e^{-x} + 1, & x < -0.1. \end{cases}$$

Notice that this stretching function is the identity function around 0, and therefore it updates $\varphi(\theta_t)$ gently when the calibration is mild ($\theta_t$ is close to 0), and faster (exponentially) as the calibration is more aggressive ($\theta_t$ is away from zero).

## C.3 THE CALIBRATION'S HYPERPARAMETERS

### C.3.1 THE BOUNDS $m, M$

the lower and upper bounds–$m$ and $M$ are predefined constants serve as safeguards against extreme situations where the data change adversarially over time. In such extreme cases, these bounds allow controlling the coverage: once $\theta$ exceeds the upper bound we return the infinite interval (the full label space) and once it exceeds the lower bound we return the empty set. By outputting the full label space, we can guarantee to control the risk at any user-specified level, as the full label space is assumed to attain loss lower than the nominal level. In practice, however, we do not expect a reasonable predictive model to reach the safeguard induced by $m$ and $M$. In fact, in our experiments, we set $m = -9999$, and $M = 9999$ to be extremely large values relative to the scale of $Y$, and the coverage we obtained is exactly 90%.

For classification problems, we can set the bounds to be $(0, 1)$, similarly to `ACI`. For regression problems, it depends on the interval constructing function $f$. If the intervals are constructed in the quantile scale, according to Section F of the main text, we can set the bounds to be $(0, 1)$ since $\theta$ is bounded in this range, as in `ACI`. If the intervals are constructed in the $Y$ scale, according to Section 3.2 of the main text, we can set them to be 100 times the difference between the lowest and highest values of the response variables in the training data.

### C.3.2 THE INITIAL VALUE OF $\theta$

The recommended way to set the initial value of $\theta$ depends on the design of the interval constructing function $f$: for example, for the interval constructing function in $Y$ scale, presented in Section 3.2 in the main text, we set the initial $\theta$ to zero as this is the right choice for a model that correctly estimates the conditional quantiles. If the model is inaccurate, $\theta$ will be updated over time, in a way that guarantees that the desired long-range coverage will be achieved.

### C.3.3 THE LEARNING RATE $\gamma$

In this section, we analyze the effect of the learning rate $\gamma$ on the performance of the calibration scheme. Figure 6 presents the results of our `Rolling RC` with a linear stretching function applied with different step-size $\gamma$ on the synthetic data described in Section C.6.1 of the main manuscript. We choose the linear stretching instead of the exponential one to better isolate the effect of $\gamma$ on the performance. Following that figure, observe that by increasing $\gamma$ we increase the adaptivity of the method to changes in the distribution of the data, as indicated by the `MSL`. Recall that (i) the lower the `MSL` the smaller the average streak of miscoverage events; and (ii) the `MSL` for the ideal model is $\approx 1.11$. On the other hand, the improvement in `MSL` comes at the cost of increasing the intervals' lengths: observe how the largest $\gamma$ results in too conservative intervals, as their `MSL` is equal to 1.

To set a proper value for $\gamma$ in regression problems, we suggest evaluating the pinball loss of the calibrated intervals, using a validation set. With this approach, one can choose the value of $\gamma$ that yields the smallest loss. We note that our method is guaranteed to attain valid coverage for any choice of $\gamma$, so the trade-off here is between the intervals' lengths and faster adaptivity to distributional shifts.

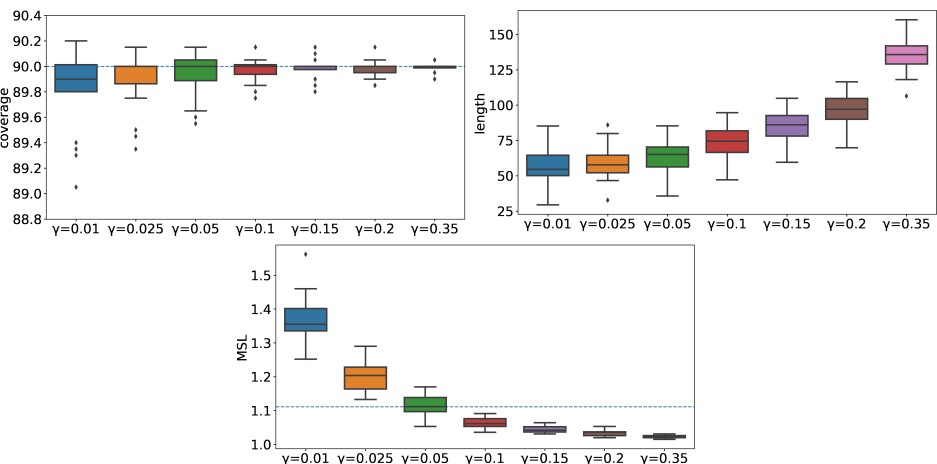

Figure 6: Rolling RC with linear stretching applied to control the 0-1 loss at level $r = 10\%$ with different learning rates on the synthetic described in Section C.6.1.

### C.4 THE HYPER-PARAMETERS FOR THE STRETCHING FUNCTIONS

The 'score adaptive' $\varphi$ function has three hyper-parameters, and the 'error adaptive' has four. Choosing them wisely greatly affects the performance of the method. To do so, we used SMAC3 (Lindauer et al., 2022), which is a hyper-parameter tuning library written in python. We let it find a combination of $\beta^{\text{score}} \in [0.01, 0.4]$, $\beta^{\text{loss}} \in [0.1, 0.2]$ that minimizes the pinball loss (or to be exact, the average of the pinball losses of the 0.95 and 0.05 quantiles), with runcount-limit of 40. We chose $\beta^{\text{low}}, \beta^{\text{high}}$ to be $-\Delta_{\text{mean}}, +\Delta_{\text{mean}}$ (respectively) where:

$$\Delta_{\text{mean}} = \frac{1}{|\mathcal{I}_{\text{val}}|} \sum_{t \in \mathcal{I}_{\text{val}}} |Y_t - Y_{t-1}|,$$

where $\mathcal{I}_{\text{val}}$ are indices $5001 - 8000$. The rationale behind this choice is that we want the clipping to be on the scale of an average change in the intervals' lengths.

### C.5 MACHINE'S SPEC

The resources used for the experiments are:

- **CPU**: Intel(R) Xeon(R) E5-2650 v4.
- **GPU**: Nvidia titanx, 1080ti, 2080ti.
- **OS**: Ubuntu 18.04.

### C.6 DATA SETS DETAILS

### C.6.1 SYNTHETIC DATA SET

In this section, we define a synthetic data set that we use in our ablation study. First, we define a group indication vector, denoted as $g_t$:

$$g = 1^{m_1} \cdot 2^{m_2} \cdot 3^{m_3} \cdot 4^{m_4}...$$

Table 1: Hyperparameters tested for each data set

| Parameter | Options |
|---|---|
| $f_1$ - LSTM input layers | [32], [32, 64], [32, 64, 128] |
| $f_2$ -LSTM layers | [64], [128] |
| $f_3$ -LSTM output layers | [32], [64, 32] |
| learning rate | $10^{-4}, 5 \cdot 10^{-4}$ |

where $w^n$ is a vector of the number $w$ repeated $n$ times, $\cdot$ is a concatenation of two vectors, $m_i \sim \mathcal{N}(500, 10^2)$ and $\mathcal{N}(\mu, \sigma^2)$ is the normal distribution with mean $\mu$ and variance $\sigma^2$. In words, each group lasts for approximately 500 time steps, and the vector is a concatenation of the group's indexes. The generation of the feature vectors and the response variable is done in the following way:

$$\hat{\beta}_i \sim \text{Uniform}(0, 1)^p,$$

$$\beta_i = \frac{\hat{\beta}_i}{\|\hat{\beta}_i\|_1},$$

$$\omega_i = \begin{cases} \mathcal{N}(20, 10), & g_t \equiv 0 \mod 2, \\ 1, & \text{otherwise,} \end{cases},$$

$$X_t \sim \text{Uniform}(0, 1)^5,$$

$$\varepsilon_t \sim \mathcal{N}(0, 1),$$

$$Y_t = \frac{1}{2}Y_{t-1} + \omega_{g_t}^2 |\beta_{g_t}^T X_t| + 2\sin(2X_{t,1} \cdot \varepsilon_t),$$

where $\text{Uniform}(a, b)$ is a uniform distribution on the interval $(a, b)$.

### C.6.2 REAL DATA SETS

In addition to the features given in the raw data set, we added for each sample the day, month, year, hours, minutes, and the day of the week. Table 2 presents the number of samples in each data set, the number of samples we used in the quantile regression experiments 4.1.2, and the dimension of the feature vector.

Table 2: Information about the real data sets.

| Data Set | Total Number of Samples | Number of Used Samples | Feature Dimension |
|---|---|---|---|
| power (Power) | 52416 | 20000 | 11 |
| energy (Energy) | 19735 | 20000 | 33 |
| traffic (Traffic) | 48204 | 20000 | 12 |
| wind (Wind) | 385565 | 20000 | 6 |
| prices (Prices) | 34895 | 20000 | 61 |

## D ADDITIONAL EXPERIMENTS

### D.1 SINGLE RESPONSE QUANTILE REGRESSION

#### D.1.1 ABLATION STUDY ON THE STRETCHING FUNCTION

In this section, we evaluate `Rolling RC` in the regression setting for different stretching functions. We follow the procedure described in Section 3.2 and use the following stretching functions:

**None.**

$$\varphi(x) = x$$

**Exponential.**

$$\varphi(x) = \begin{cases} e^x - 1, & x > 0, \\ -e^{-x} + 1, & x \leq 0, \end{cases}$$

**Score adaptive.** The following stretching function makes a larger update to $\theta_t$ the farther the test $Y_t$ from the interval's boundaries. This is in contrast with the exponential function described above, which does not take into account the quality of the constructed interval. More formally, denote the CQR non-conformity score (Romano et al., 2019) by

$$s_t = \max\{\mathcal{M}_t(X_t, \alpha/2) - Y_t, Y_t - \mathcal{M}_t(X_t, 1 - \alpha/2)\},$$

which measures the signed distance of $Y_t$ from the its closest boundary. Next, define

$$\varphi_t(\theta) = \theta + \lambda_t^{\text{score}}, \quad \text{where} \quad \lambda_t^{\text{score}} = \text{clip}(\lambda_{t-1}^{\text{score}} - \beta^{\text{score}} \cdot s_{t-1}, \beta^{\text{low}}, \beta^{\text{high}}),$$

where $\beta^{\text{score}}$, $\beta^{\text{low}}$ and $\beta^{\text{high}}$ are hyperparameters. Similarly to the 'error adaptive' stretching function presented in Section 3.2.1, the clipping function is used to restrain the effect of an outlier $Y_t$ that is far from the boundaries.

**Error adaptive.** By adding awareness of previous points' loss to the 'score adaptive' stretching, we forge a stretching function that is aware of both the error margin and the constructed intervals' loss:

$$\varphi_t(\theta) = \theta + \lambda_t^{\text{error}}, \quad \text{where} \ \lambda_t^{\text{error}} = \text{clip}(\lambda_{t-1}^{\text{error}} - \beta^{\text{score}} \cdot s_{t-1} \cdot \exp\left\{\beta^{\text{loss}} \cdot |\ell_{t-1} - r|\right\}, \beta^{\text{low}}, \beta^{\text{high}}).$$

Figure 7 displays the performance of `Rolling RC` aiming to control coverage rate at level $1 - \alpha = 90\%$ with the stretching functions described above, and Figure 8 presents the results of `Rolling RC` applied to control the `MC` risk at level $\alpha/(1 - \alpha) = 1/9$. Following these figures, we can see that `Rolling RC` with each stretching function performs well on most of the metrics on most of the data sets.

Although the 'no stretching' and the 'exponential stretching' functions converge faster to the desired risk level, it is clear from the results that the 'score adaptive' and the 'error adaptive' stretching functions construct narrower intervals. Moreover, the 'error adaptive' approach is superior in several terms:

- It constructs the shortest intervals.

- It achieves `MSL` that is closer to the ideal level 1.111..., which means that consecutive miscoverage events are less likely to occur (see Appendix E.1).

- It achieves `MC` that is closer to the desired level 0.111... when aiming to control the coverage rate, and its coverage rate is closer to $90\%$ when aiming to control the `MC` risk level, which is a desired outcome (see Appendix E.2).

- `Rolling RC` with 'error adaptive' stretching performs similarly to the competitive stretchings in terms of $\Delta$`Coverage`.

### D.1.2 CONSTRUCTING UNCERTAINTY SETS WITH A CALIBRATION SET

In this section, we analyze an instantiation of `ACI` (Gibbs & Candes, 2021), which we refer to as `calibration with cal` that constructs uncertainty sets with a controlled miscoverage rate using a calibration set. It is more out-of-the-box because it allows the user to take any conformal score function from the conformal prediction literature to get a confidence set function $f$. Many conformal scores have been developed and extensively studied, so this approach directly inherits all of the progress made on this topic. `calibration with cal` uses calibration points, but does not hold out a large block. Rather, previous points are simultaneously used both for calibration and model fitting.

Turning to the details, denote by $S(\mathcal{M}_t(X_t), Y_t) \in \mathbb{R}$ a non-conformity score function that takes as an input the model's prediction $\mathcal{M}_t(X_t)$ at time $t$ and the corresponding label $Y_t$, and returns a measure for the model's goodness-of-fit or prediction error. Here, the convention is that smaller scores imply a better fit. For instance, adopting the same notations from (6), the quantile regression

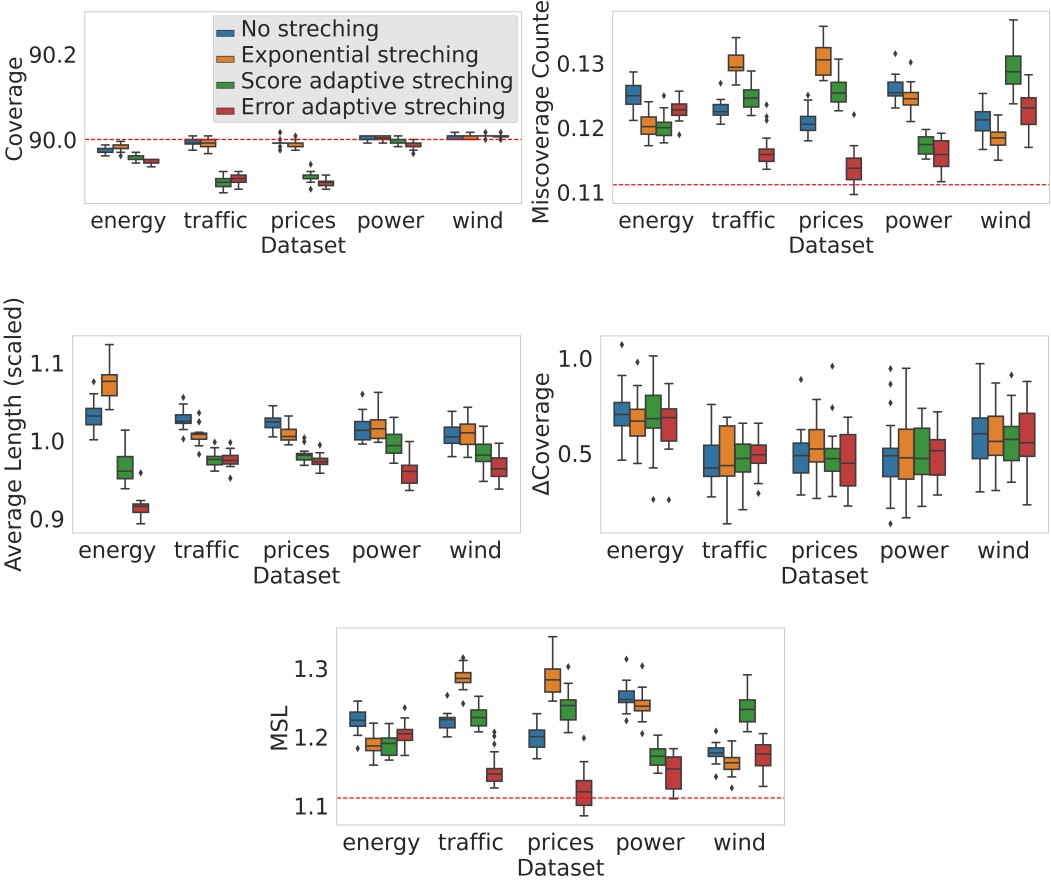

Figure 7: Performance of `Rolling RC` on real data sets, aiming to control the coverage rate at level $1 - \alpha = 90\%$. The length of the prediction intervals is scaled per data set by the average length of the constructed intervals. Results are evaluated on 20 random initializations of the predictive model. The $\Delta$`Coverage` metric is scaled between 0 to 100.

score presented in (Romano et al., 2019) are given by $S(\mathcal{M}_t(X_t), Y_t) = \max\{\mathcal{M}_t(X_t, \alpha/2) - Y_t, Y_t - \mathcal{M}_t(X_t, 1 - \alpha/2)\}$. Next, define the prediction set constructing function in (4) as:

$$f(X_t, \theta_t, \mathcal{M}_t) = \{y \in \mathcal{Y} : S(\mathcal{M}_t(X_t), y) \le Q_{1+\theta_t}(\mathcal{S}_{\text{cal}})\}, \quad (12)$$

where $\mathcal{S}_{\text{cal}} = \{S(M_{t'}(X_{t'}), Y_{t'}) : t' = t - n, \ldots, t - 1\}$ is a set containing the $n$ most recent non-conformity scores. The function $Q_{1+\theta_t}(\mathcal{S}_{\text{cal}})$ returns the $(1 + \theta_t)$-th empirical quantile of the scores in $\mathcal{S}_{\text{cal}}$, being the $\lceil (1 + \theta_t)(n + 1) \rceil$ largest element in that set. Here, $-1 \le \theta_t \le 0$ is the calibration parameter we tune recursively, as in (5). The reason for having the negative sign, is to form larger prediction sets as $\theta$ increases. In plain words, $f$ in (12) returns all the candidate target values $y$ for the test label, whose score $S(\mathcal{M}_t(X_t), y)$ is smaller than $(1 + \theta_t) \times 100\%$ of the scores in $\mathcal{S}_{\text{cal}}$, which are evaluated on truly labeled historical data $S(\mathcal{M}_{t'}(X_{t'}), Y_{t'})$. As such, the size of the set in (12) gets smaller (larger) as $1 + \theta_t$ gets smaller (larger).

For reference, `calibration with cal` procedure is summarized in Algorithm 2. The reason for this method's name is to emphasize that we now use calibration scores to formulate the prediction set function $f$. In fact, the coverage guarantee of `calibration with cal` follows directly from Theorem 1 for $f(X_t, \theta_t, \mathcal{M}_t)$ defined in (12).

We run `Rolling RC` with either 'error adaptive' stretching and without stretching, as described in Section 3.2.1, and `calibration with cal`, as presented in Algorithm 2 on the real data sets detailed in Appendix C.6.2. Figure 9 summarizes the results, showing that all methods attain the desired coverage level; this is guaranteed by Theorem 1. This figure also shows that `Rolling RC`

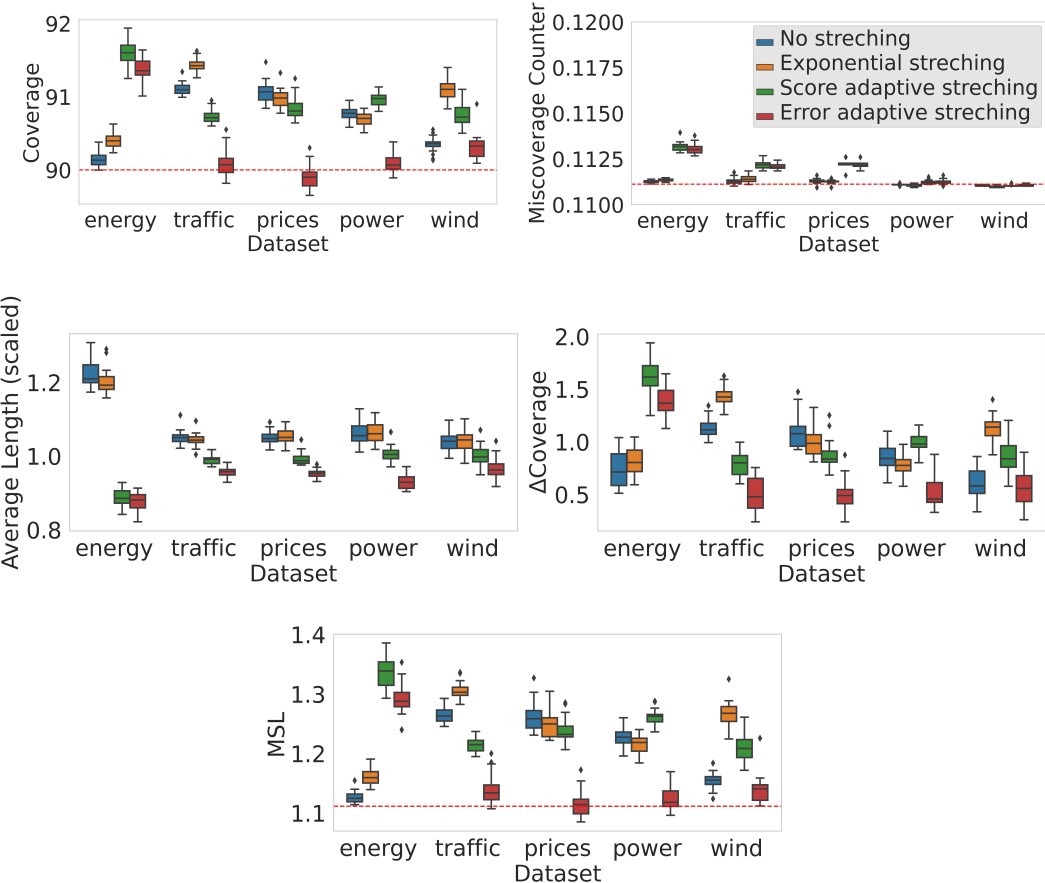

Figure 8: Performance of `Rolling RC` on real data sets, where we aim to control the `MC` risk at level $\alpha/(1-\alpha) = 1/9$. The length of the prediction intervals is scaled per data set by the average length of the constructed intervals. Results are evaluated on 20 random initializations of the predictive model. The $\Delta$`Coverage` metric is scaled between 0 to 100.

with 'error adaptive' stretching constructs the narrowest intervals while attaining the best conditional coverage metrics. Furthermore, one can see that even without stretching, `Rolling RC` performs better than `calibration with cal`, as indicated by the intervals' lengths and the conditional coverage metrics.

## D.2 UNCERTAINTY QUANTIFICATION FOR ONLINE DEPTH ESTIMATION

In this section we analyze the performance of the uncertainty quantification heuristics described in Section 3.2.1. We follow the experimental protocol explained in Section C.2.4, and display the results in Figure 10. This figure shows that all heuristics attain the nominal image coverage level, as guaranteed by Theorem 1. Furthermore, the figure suggests that estimating the current residual with the five most recent ones outperforms the baseline constant technique, while estimating the residual with a neural network, does not, as indicated by the average length and center coverage metrics. We propose two possible explanations for this phenomenon. First, since the residual model's architecture is huge, it may require further offline fitting, on a larger data set. For comparison, we trained it for 60 epochs on 6000 samples, while the base depth prediction model, LeReS Yin et al. (2021), was trained over 300k samples. Second, since one can extract a depth estimate from a residual estimate, the problem of estimating the residual $r$ is equivalent to estimating the depth:

$$\hat{Y}_t = r(X_t) + \mathcal{M}(X_t).$$

---

**Algorithm 2** `calibration with cal`

---

**Input:**

Data $\{(X_t, Y_t)\}_{t=1}^T \subseteq \mathcal{X} \times \mathcal{Y}$, given as a stream, miscoverage level $\alpha \in (0, 1)$, a score function $S$, a calibration set size $n_2$, a step size $\gamma > 0$, and an online learning model $\mathcal{M}$.

**Process:**

1: Initialize $\alpha_0 = \alpha$ and a set of the previous conformity scores: $\mathcal{S}_{\text{cal}} = \emptyset$.
2: **for** $t = 1, ..., T$ **do**
3:     Construct a prediction set for the new point $X_t$:

$$\hat{C}_t^{\text{WC}}(X_t) = \{y \in \mathcal{Y} : S(\mathcal{M}_t(X_t), y) \leq Q_{1-\alpha_t}(\mathcal{S}_{\text{cal}})\}.$$

4:     Obtain $Y_t$.
5:     Compute the current conformity score: $s_t = S(\mathcal{M}_t(X_t), Y_t)$.
6:     Add the current conformity score to the set: $\mathcal{S}_{\text{cal}} = \mathcal{S}_{\text{cal}} \cup \{s_t\}$.
7:     Remove the oldest calibration point from the set: $\mathcal{S}_{\text{cal}} = \mathcal{S}_{\text{cal}} - \{s_{t-n_2}\}$.
8:     Compute $\text{err}_t = \mathbb{1}\{Y_t \notin \hat{C}_t^{\text{WC}}(X_t)\}$.
9:     Update $\alpha_{t+1} = \alpha_t + \gamma(\alpha - \text{err}_t)$.
10:     Fit the model $\mathcal{M}_t$ on $(X_t, Y_t)$ and obtain the updated model $\mathcal{M}_{t+1}$.
11: **end for**

**Output:**

Uncertainty sets $\hat{C}_t^{\text{WC}}(X_t)$ for each time step $t \in \{1, ..., T\}$.

---

Therefore, the fact that estimating a depth map is extremely difficult, as explained in Section C.2.6, turns the task of estimating the residual to be difficult as well. As a consequence, the residual estimates may be inaccurate.

### D.3 MULTIPLE RISKS CONTROL: ANALYZING THE AGGREGATION FUNCTIONS

In this section we examine two options for aggregating the vector $\underline{\theta}_t$ into a scalar through $\lambda_t$:

**Mean.** $\lambda_t^{\text{mean}} = \frac{1}{2}(\varphi(\underline{\theta}_t^1) + \varphi(\underline{\theta}_t^2))$. Taking the average of the entries compromises between the different risks and results in intervals that are not too conservative and not too liberal.

**Max.** $\lambda_t^{\text{max}} = \max\{\varphi(\underline{\theta}_t^1), \varphi(\underline{\theta}_t^2)\}$. Since the maximal coordinate corresponds to the most conservative loss, the constructed intervals may be too conservative.

The third possible aggregation is the minimum function $\lambda_t^{\text{min}} = \min(\underline{\theta}_t)$ that consistently follows the minimal entry in $\underline{\theta}_t$. Since the minimal coordinate in $\underline{\theta}_t$ corresponds to the most liberal loss, this approach is likely to result in intervals that are too liberal, as it ignores the conservative losses. Therefore, we do not examine this aggregation in our experiments.

We follow the experimental setup described in Section 4.2 and display in Figure 11 the performance of the mean and max aggregation for $\underline{\theta}_t$. This figure shows that the two methods perform similarly, and using mean aggregation leads to slightly narrower intervals compared to the mean aggregation approach.

### E  TIME-SERIES CONDITIONAL COVERAGE METRICS

#### E.1  AVERAGE MISCOVERAGE STREAK LENGTH

Following Section 4.1.1, recall that the miscoverage streak length of a series of intervals $\{\hat{C}_t(X_t)\}_{T_0}^{T_1}$ is defined as:

$$\text{MSL} := \frac{1}{|\mathcal{I}|} \sum_{t \in \mathcal{I}} \min\{i : Y_{t+i} \in \hat{C}_{t+i}(X_{t+i}) \text{ or } t = T_1\},$$

where $\mathcal{I}$ is a set containing the starting times of all miscoverage streaks:

$$\mathcal{I} = \left\{t \in [T_0, T_1] : \left(t = T_0 \text{ or } Y_{t-1} \in \hat{C}_{t-1}(x_{t-1})\right) \text{ and } Y_t \notin \hat{C}_t(X_t)\right\}.$$

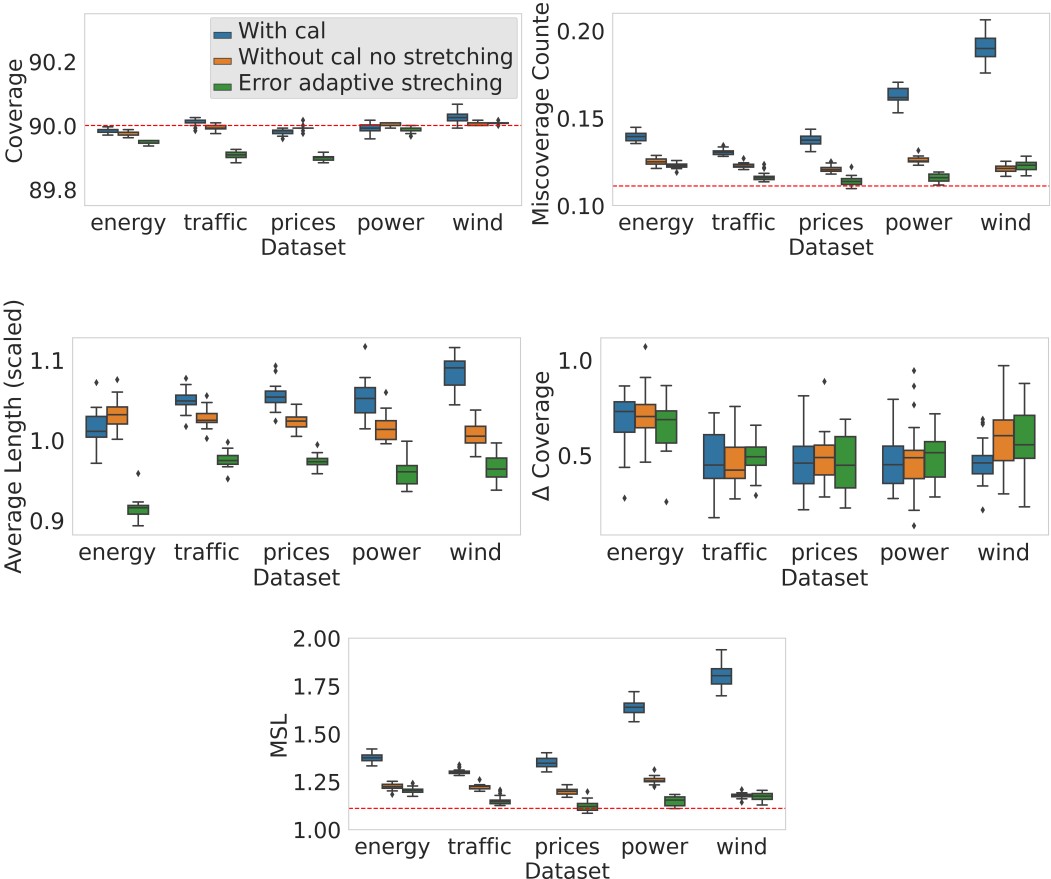

Figure 9: Performance of `calibration with cal` (Algorithm 2) (blue), `Rolling RC` without stretching (orange), and `Rolling RC` with 'error adaptive' stretching (green). Results are evaluated on 20 random initializations of the predictive model. The $\Delta$`Coverage` metric is scaled between 0 to 100.

Above, $[T_0, T_1]$ is the set of all integers between $T_0$ and $T_1$.

To clarify this definition of the `MSL`, we now analyze the `MSL` in two concrete examples. Denote by "1" a coverage event and by "0" a miscoverage event, and consider a sequence of 15 observations. A method that results in the following coverage sequence:

$$1, 1, 1, 1, 1, 1, \mathbf{0}, 1, \mathbf{0,0}, 1, 1, 1, 1, 1,$$

has an `MSL` $= (2+1)/2 = 1.5$, and coverage $= 12/15 = 80\%$. By contrast, a method that results in the following sequence

$$1, 1, 1, 1, 1, 1, 1, 1, 1, \mathbf{0,0,0},$$

has the same average coverage of $80\%$ but much larger `MSL` $= 3/1 = 3$. This emphasizes the role of `MSL`: while the two methods cover the response in 12 out of 15 events, the second is inferior as it has, on average, longer streaks of miscoverage events.

We now compute the `MSL` of intervals constructed by the true conditional quantiles $\{C(X_t)\}_{t=T_0}^{T_1}$. By construction, these intervals satisfy:

$$\mathbb{P}(Y_t \in C(X_t) \mid X_t = x_t) = 1 - \alpha.$$

Therefore, $Z_t = \min\{i : y_{t+i} \in \hat{C}(X_{t+i}) \text{ or } t = T_1\}$ is a geometric random variable with success probability $1 - \alpha$. The average miscoverage streak length of the true quantiles is the mean of $Z_t$,

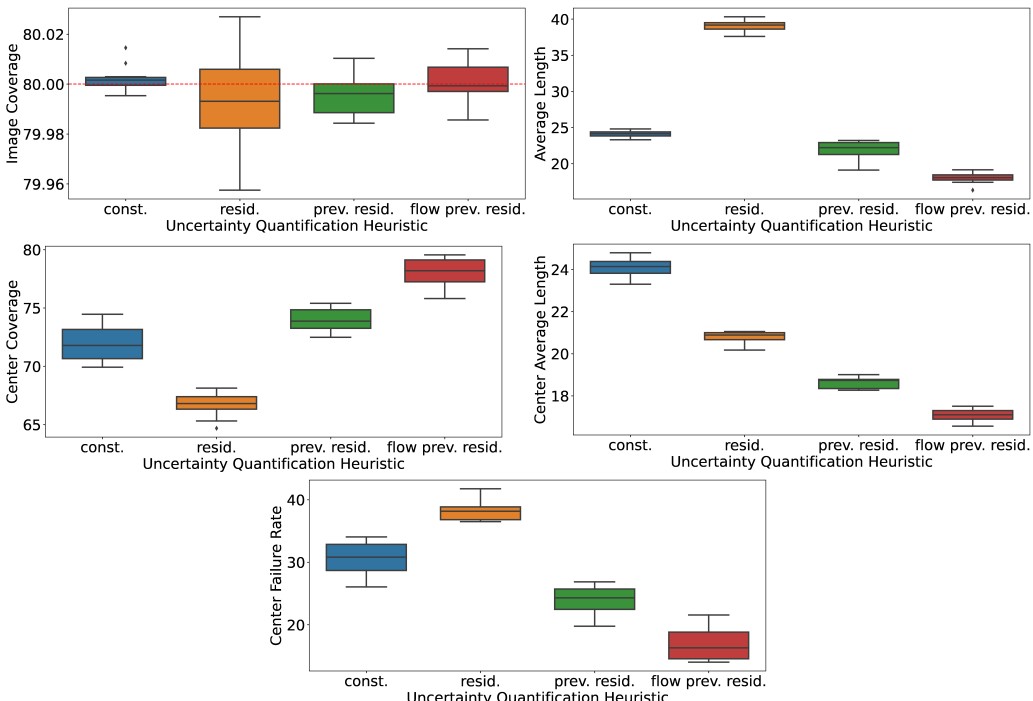

Figure 10: Performance of `Rolling RC` applied to control the image coverage rate with the following uncertainty quantification heuristics that are described in Section B.2: 'constant value' (blue), 'magnitude of the residual' (orange), 'previous residuals' (green) and 'previous residuals with optical flow registration' (red). All methods use the exponential stretching function introduced in Section 3.2.1.

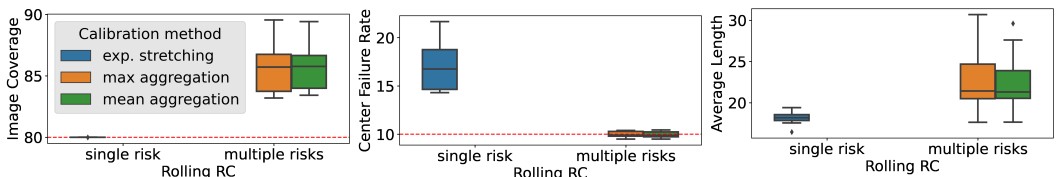

Figure 11: Performance of `Rolling RC` applied to control both 'image miscoverage' and 'center failure' risks. All methods use the exponential stretching function introduced in Section 3.2.1.

which is:

$$\texttt{MSL} = \frac{1}{1 - \alpha} \underset{\alpha = 0.1}{\approx} 1.111.$$

Therefore, having $\texttt{MSL} = 1$ is not necessarily equivalent to an optimal conditional coverage, as it indicates for undesired anti-correlation between two consecutive time steps: after a miscoverage event follows a coverage event with probability one. Consequently, we would desire to have $\texttt{MSL} = \frac{1}{1-\alpha}$, which is the $\texttt{MSL}$ attained by the true conditional quantiles.

### E.2 THE MISCOVERAGE COUNTER

How should one choose the risk level for the `MC` risk? If we aim at $1 - \alpha = 90\%$ coverage, we argue that the right choice is $r = \alpha/(1 - \alpha) = 1/9$. To see this, suppose we have an ideal model that attains a perfect coverage rate $1 - \alpha$ conditional on $t$. In this case, the coverage events are i.i.d. realizations of Bernoulli experiments and so `MC` acts as a geometric random variable that counts the number of failures until reaching a success, where the success probability is $1 - \alpha$. Hence, we define the `MC` risk level $r$ to be the expected value of such a geometric random variable, which is $\alpha/(1 - \alpha)$. By Proposition 1 we know that if we control `MC` at level $r = 1/9$, the coverage will be

at least $1 - \alpha \approx 88.89\%$, where an ideal model will reach an exact $90\%$ coverage. This stands in striking contrast with a method that only controls the coverage metric, as the constructed prediction intervals may result in a large MC risk.

Note that Theorem 1 assumes that the loss is bounded, while MC is not bounded. To guarantee that `Rolling RC` will converge to the desired risk level of MC, we can use the following loss instead: $\text{MC}' = \min\{\text{MC}, B\}$ for some large $B \in \mathbb{N}$. In the experiments, however, we used the regular MC as we observed that its value does not get too high in practice.

### E.3  $\Delta$COVERAGE

The time-series data sets we use in the experiments in Section 4.1.2 include the day of the week as an element in the feature vector. Therefore, we assess the violation of day-stratified coverage (Feldman et al., 2021; Zaffran et al., 2022), as a proxy for conditional coverage. That is, we measure the average deviation of the coverage on each day of the week from the nominal coverage level. Formally, given a series of intervals $\{\hat{C}_t(X_t)\}_{T_0}^{T_1}$, their $\Delta$Coverage is defined as:

$$\Delta\text{Coverage} = \frac{1}{7} \sum_{i \in \{1,2,\dots,7\}} \left| \frac{1}{|D_i|} \sum_{t \in D_i} \mathbb{1}_{Y_t \in \hat{C}_t(X_t)} - (1 - \alpha) \right|,$$

where $D_i$ is a set of samples that belong to the $i$-th day of the week. Since a lower value of this metric indicates for a better conditional coverage, we desire to have a minimal $\Delta$Coverage.

## F  CALIBRATING ON THE QUANTILE SCALE IN REGRESSION TASKS

As an alternative for (6), where the calibration coefficient $\varphi(\theta_t)$ is added to each of the interval endpoints, one can modify the interval's length by tuning the raw miscoverage level $\tau_t = \varphi(\theta_t)$ requested from the model:

$$f(X_t, \theta_t, \mathcal{M}_t) = [\mathcal{M}_t(X_t, \tau_t/2), \ \mathcal{M}_t(X_t, 1 - \tau_t/2)].$$

This formulation is inspired by the work of (Chernozhukov et al., 2021) that suggested tuning the nominal miscoverage level $\tau$, based on a calibration set. In contrast to (6), where we estimate only the lower $\alpha/2$ and upper $1 - \alpha/2$ conditional quantiles, here, we need to estimate all the quantiles simultaneously. To accomplish this, one can apply the methods proposed in (Chung et al., 2021; Park et al., 2021; Sesia & Romano, 2021). Turning to the choice of the stretching function $\varphi$: the straightforward option is to set $\varphi(\theta) = -\tau$, where $\theta_t$ is bounded in the range: $-1 \leq \theta_t \leq 0$. The reason for having the negative sign in $\varphi$, is to form larger prediction sets (resulted by smaller values of $\tau$) as $\theta$ increases.

## G  CONSTRUCTING PREDICTION SETS FOR CLASSIFICATION TASKS

Consider a multi-class classification problem, where the target variable is discrete and unordered $y \in \mathcal{Y} = \{1, 2, \dots, K\}$. Suppose we are handed a classifier that estimates the conditional probability of $P_{Y_t|X_t}(Y_t = y \mid X_t = x)$ for each class $y$, i.e., $\mathcal{M}_t(X_t, y) \in [0, 1]$ and $\sum_{y \in \mathcal{Y}} \mathcal{M}_t(X_t, y) = 1$. With this in place, we follow (Papadopoulos et al., 2002) and define the prediction set constructing function as:

$$f(X_t, \theta_t, \mathcal{M}_t) = \{y : \mathcal{M}_t(X_t, y) \geq \varphi(\theta_t)\}, \tag{13}$$

where one can choose $\varphi(x) = -x$, for instance. While this procedure is guaranteed to attain the pre-specified risk level $r$, according to Theorem 1, the function $f$ in (13) may have unbalanced coverage across different sub-populations in the data (Angelopoulos et al., 2021b; Cauchois et al., 2021). To overcome this, we recommend using the function $f$ presented next, which is capable of constructing prediction sets that better adapt to the underlying uncertainty. The idea, inspired by the work of (Angelopoulos et al., 2021b; Romano et al., 2020), is to initialize an empty prediction set and add class labels to it, ordered by scores produced by the model. We keep adding class labels until the total score exceeds $1 - \alpha$. Formally, the confidence set function is defined as:

$$f(X_t, \theta_t, \mathcal{M}_t) = \{\pi_1, \dots, \pi_k\}, \text{ where } k = \inf\left\{k : \sum_{j=1}^{k}(\mathcal{M}_t(X_t, \pi_j)) \geq 1 - \varphi(\theta_t)\right\},$$

and $\pi$ is the permutation of $\{1, 2, ...K\}$ sorted by the scores $\{\mathcal{M}_t(X_t, y) : t \in \mathcal{Y}\}$ from the highest to lowest. As for the stretching function $\varphi$, we recommend using $\varphi(x) = x$.

