# OpenReview forum: "Risk Control for Online Learning Models"
_ICLR.cc/2023/Conference — Submitted to ICLR 2023_

### Official Review · Reviewer_sghz · 2022-10-19

**Confidence:** 4
**Correctness:** 3
**Technical Novelty And Significance:** 2
**Empirical Novelty And Significance:** 2
**Recommendation:** 6

**Clarity, Quality, Novelty And Reproducibility:**

**Clarity**: The paper is clear. Minor comments:
- I cannot see color encoding in the first figure in Figure 3; probably rescaling the x-axis addresses this issue.
- In Line 1 in Algorithm 1, I guess theta_0 is initialized by m instead of 0.
- The Figure 5 caption (“image miscoverage”) and the axis label (“image coverage”) mismatch makes reading very confusing.

**Quality**: The claims seem to be theoretically and empirically supported.

**Novelty**: The claimed novelty is weak. For the validity, Rolling RC supports general risk functions, but its usefulness is not convincing (as mentioned in weaknesses). For the compatibility of the approach, this is not a unique contribution of this paper. For the fast reaction to distribution shift, it is useful under the iid assumption, but not fully convinced its efficacy in distribution shift (as mentioned in weaknesses)

**Reproducibility**: The algorithm is simple, so I believe that this work is reproducible on evaluated cases. Hyperparameter tuning is tricky, but the paper provides some guidelines.


**Strength And Weaknesses:**

**Strengths**:
* Rolling RC is simpler than ACI
* Rolling RC can control any risk, which is novel in online settings

**Weaknesses**:
* **W1**. Depth estimation use-case is not convincing
* **W2**. The claimed contributions are weak.
* **W3**. The convergence bound is not convincing (in an adversarial sequence)
* **W4**. Weak or no comparison in related work or results

**Discussion**

**W1**. In Section 1.1, the paper introduces online depth estimation as a motivational example for justifying risk controlling. Here, the paper considers (I guess) depth measurements from accurate sensors (e.g., Lidar) as the ground truth depth, and claimed the necessity for uncertainty quantification for speed-up depth sensing. I think this may be a good example, but there are many missing pieces. First, I’m not sure if there are actual benefits in speed-up by using depth estimation via RGB instead of Lidar. In close range, Lidar can collect accurate depth very quickly, but for far objects, anyway we don’t need accurate depth, for example, in self-driving scenarios. Secondly, this motivation example uses the sensor measurements from an accurate sensor as ground truth, but it is unrealistic. For example, Lidar can be very noisy or wrong in snowy weather or puddles in the ground. So, it provides wrong labels to online learning algorithms. At least these two points undermine the justification on the necessity of risk controlling in online settings. Could you refine the motivation example more?

**W2**. The paper claims three contributions: (1) validity on the risk of constructed prediction sets, (2) compatibility with online learning methods, and (3) fast reaction to distribution shift.

For the risk validity, it is novel in online settings (considering risk controlling is possible under iid), but I’m not convinced why we need risk controlling as mentioned in W1.

For the second contribution, I think this holds for any conformal approaches, including ACI; I don’t think this is a unique contribution of this paper.

For the fast reaction to the distribution shift, I feel that this is a bit contradictory. The paper proposes several interesting stretch functions to reduce prediction set size, while maintaining a desired coverage. But, choosing a stretch function is equivalent to hyperparameter tuning. Meaning that, a good stretch function depends on the property of a data sequence, but simultaneously online algorithms, including Rolling RC, want to work for any distributions. I think I mainly get this impression on the way of choosing hyperparameters of this paper, i.e., hold out an initial subsequence in the entire sequence and test on the later subsequence. This hold-out-first-sequence approach does not make sense in distribution shift, as the initial subsequence cannot represent the later sequence. Could you justify this tuning approach?


**W3**. I was trying to understand the convergence bound in the last equation of Theorem 1 proof. Mainly, I was curious about why this deterministic algorithm could work in adversarial settings; recall that traditional online learning algorithms are randomized such that an adversary could not entirely fool the algorithms (or need a realisability assumption, e.g., for the perceptron algorithm).

To make the analysis simple, let m=0, M=1, \gamma=1, \theta_1=0, risk is the indicator loss, and r=0.1. Then, this convergence bound means the absolute difference of the average coverage rate and a desired coverage rate r is bounded by 1/T. By setting T = 10, this means after 10 time steps, the average coverage rate is at most 0.2 for any sequences. So, an adversary can choose an odd-subsequence (i.e., data arrive in odd time) such that a prediction set makes an error. Then, the \theta changes as follows:

t=1: 0 -> 0.9,
t=2: 0.9 -> 0.8,
t=3: 0.8 -> 1,
t=4: 1 -> 0.9,
t=5: 0.9 -> 1,
…
As can be seen, at the odd time, \theta is not one, the adversary can generate a datum to make an error. This means the average coverage rate is 0.5, which is larger than the bound by theorem. Could you correct my understanding? If not, what are additional assumptions to achieve the desired coverage rate?

**W4**. The risk controlling aspect is new in online learning, but I’m not sure why we cannot use the risk controlling idea along with ACI? As mentioned, ACI updates the coverage rate, while Rolling RC updates the threshold directly. What is the reason that we cannot control risk in ACI? If that’s possible, what’s the benefit of using risk controlling in threshold update? This may be justified empirically.



**Summary Of The Paper:**

Uncertainty quantification is necessary in high-stakes applications, where both high predictive accuracy and reliable safeguards to handle unanticipated changes in data generation. This paper addresses the uncertainty quantification in online settings based on recent advances in adaptive conformal prediction. The proposed approach, called Rolling RC, updates a prediction set as data arrive sequentially, while achieving a desired coverage. The proposed approach is evaluated on five real-world datasets for one-dimensional response and one depth prediction dataset for high-dimensional response, showing that the proposed approach roughly achieves the desired coverage. The claimed contributions include (1) validity on the risk of constructed prediction sets, (2) compatibility with online learning methods, and (3) fast reaction to distribution shift.

**Summary Of The Review:**

I think this paper attacks an interesting online conformal prediction by extending ACI for risk controlling, but currently, weaknesses overweigh strengths. So, I lean to rejection, but willing to adjust my understanding.


==== post rebuttal

Thanks for the response. My main concern was addressed so I raised my score. The following includes remaining minor concerns.

* I agree that online depth estimation is one showcase of online risk controlling, but I'm still not well-motivated whether we need UQ in online depth estimation as I initially mentioned. I hope someone else come up with a cool application of risk controlling Rolling-RC
* In comparison to ACI, I initially expected that ACI and Rolling-RC would have similar performance in the standard coverage rate (i.e., the binary loss), but it is not as shown in Figure 9. It would be interesting to discuss the performance difference (e.g., “calibration with cal” is sensitive to the calibration set size).

---

> ### Author Response · Authors · 2022-11-17
> **References**
>
>
> [1] Cho, Younggun, Young-Sik Shin, and Ayoung Kim. "Online depth estimation and application to underwater image dehazing." OCEANS 2016 MTS/IEEE Monterey. IEEE, 2016.
>
> [2] Zhang, Zhenyu, Stephane Lathuiliere, Elisa Ricci, Nicu Sebe, Yan Yan, and Jian Yang. "Online depth learning against forgetting in monocular videos." In Proceedings of the IEEE/CVF Conference on Computer Vision and Pattern Recognition, pp. 4494-4503. 2020.
>
> [3] Maslov, Dmitrii, and Ilya Makarov. "Online supervised attention-based recurrent depth estimation from monocular video." PeerJ Computer Science 6 (2020): e317.
>
> [4] Patil, Vaishakh, Wouter Van Gansbeke, Dengxin Dai, and Luc Van Gool. "Don’t forget the past: Recurrent depth estimation from monocular video." IEEE Robotics and Automation Letters 5, no. 4 (2020): 6813-6820.
>
> [5] Klingner, Marvin, and Tim Fingscheidt. "Online performance prediction of perception DNNs by multi-task learning with depth estimation." IEEE Transactions on Intelligent Transportation Systems 22.7 (2021): 4670-4683.
>
> [6] Jeon, Ikgeun, Liu Yang, Kwangnam Ryu, and Hoon Sohn. "Online melt pool depth estimation during directed energy deposition using coaxial infrared camera, laser line scanner, and artificial neural network." Additive Manufacturing 47 (2021): 102295.
>
> [7] Bates, Stephen, et al. "Distribution-free, risk-controlling prediction sets." Journal of the ACM (JACM) 68.6 (2021): 1-34.‏
>
> [8] Angelopoulos, Anastasios N., et al. "Learn then test: Calibrating predictive algorithms to achieve risk control." arXiv preprint arXiv:2110.01052 (2021).‏
>
> [9]  Angelopoulos, Anastasios N., et al. "Conformal risk control." arXiv preprint arXiv:2208.02814 (2022).‏
>
> [10] Isaac Gibbs and Emmanuel Candès. Adaptive conformal inference under distribution shift. In Advances in Neural Information Processing Systems, 2021.

---

> ### Author Response · Authors · 2022-11-17
> **Reply to the review (2)**
>
> ## Compatibility with online learning methods
> We thank the reviewer for the opportunity to expand on our contributions. Indeed, offline conformal methods are compatible with offline learning methods. However, in online settings, it quickly becomes infeasible to deploy an offline learning algorithm whose training is computationally expensive. This would result in a dramatic delay in the prediction process. However, our $\texttt{Rolling RC}$ is cheaper to apply since it adds negligible computational costs to any online learning algorithm---whose training is done online, in a computationally efficient manner. This stands in contrast with Algorithm 1 in [10] ($\texttt{ACI}$), for example, which requires refitting an entire model and computing the conformity score on each of the calibration samples at every time step. For these reasons, we find the compatibility with online learning methods and efficiency of $\texttt{Rolling RC}$ as significant properties, as they are not common among existing methods. In response to this comment, we updated the contribution section to reflect better these points.
>
>
> ## Applying a stretching function is more than hyperparameter tuning
>
> Excellent point! Figure 7 in Appendix D.1.1 compares $\texttt{Rolling RC}$ applied with different stretching functions. In this experiment, the calibration's updating rate parameter, $\gamma$, is chosen to obtain the best performance on a validation set. That is, the hyperparameters are tuned over a validation set for all stretching functions. This figure shows that $\texttt{Rolling RC}$ applied with any stretching function outperforms $\texttt{Rolling RC}$ without stretching. Therefore, we truly believe that the stretching function is important for improving statistical efficiency.
>
> ## Hyperparameter tuning based on a validation set
> We thank the reviewer for this important comment.
> Indeed, using a validation set to tune the hyperparameters and the choice of the stretching function are heuristics. Yet, we found this rule of thumb to work well empirically, as visualized in Figure 7 in Appendix D.1.1. Of course, there are situations where this approach would not be most effective since the data is not i.i.d., yet we believe that our suggestion is sensible as clearly indicated by our experiments. Another advantage of tuning the hyperparameters this way is that it facilitates the comparison between the different methods since they all follow the same automatic hyperparameter tuning approach. We added this discussion to the revised manuscript in Appendix C.1.3.
>
> We thank the reviewer for this comment, which greatly improves the clarity of our paper!
> ## The convergence bound is not convincing
> We thank the reviewer for bringing up this valuable subject. As the reviewer suggests,  Theorem 1 states that the average coverage rate falls in the range $[0.7,1]$. The ``bug'' in the provided example is that $\texttt{Rolling RC}$ outputs the full label set, i.e., $C(X_t)=\mathcal{Y}$, for timestamps $t$ for which $\theta_t>M$. By outputting the full label space, we can provably control the risk (coverage rate, in this case) at any user-specified level. Additionally, $\texttt{Rolling RC}$ does not clip $\theta$ when it exceeds the range $[m,M]$. Therefore, at $t=3$ we get $\theta=0.8+0.9=1.7$, and at $t=4$ $\texttt{Rolling RC}$ outputs the full label set which certainly covers $Y_t$.
>
> We stress that the bounds $m,M$ serve as safeguards in adversarial cases and are usually irrelevant in practice. For instance, in all experiments provided in Section 4 we set $m=-9999, M=9999$, so that $\theta$ never exceeds these values. Yet, we obtain valid risk even though $\texttt{Rolling RC}$ did not output the full label space in any of the experiments.
> We clarified this point in the revised manuscript in Appendix C.3.1.
>
>
>
>
> ## Minor comments
> We thank the reviewer for raising these problems. We updated the manuscript accordingly.

---

> ### Author Response · Authors · 2022-11-17
> **Reply to the Review (1)**
>
> We very much appreciate your positive feedback and interest in our work. We thank the reviewer for classifying our contribution as a novel one. We also thank the reviewer for their valuable feedback and suggestions. In what follows, we address your concerns in detail.
>
> ## The novelty of $\texttt{Rolling RC}$ -- $\texttt{ACI}$ cannot be used to control risks.
>
> Thank you for this valuable comment which was raised by the other reviewers, and addressed in the reply to all reviewers. In short, one cannot naively plug an arbitrary loss function into $\texttt{ACI}$ and achieve risk control. The reason is that $\texttt{ACI}$ works with conformity scores that are only relevant to the miscoverage loss, but do not exist in the general risk-controlling setting. We updated the introduction to better clarify this point.
> For the convenience of the reviewer, we repeat our answer below.
>
> We agree that from a theoretical perspective we rely on the mathematical foundations of $\texttt{ACI}$, but we do not see this as a limitation, as our contribution is practical rather than technical. In this work, we show how to broaden the set of problems that $\texttt{ACI}$ can tackle and control an arbitrary loss, which is far from being straightforward. To put it simply, a reader of $\texttt{ACI}$ would not know how to use this technique in online risk-controlling tasks, such as the depth sensing example. Specifically, one cannot naively plug an arbitrary loss function into $\texttt{ACI}$ to achieve risk control. The reason is that $\texttt{ACI}$ works with conformity scores that are relevant only to the miscoverage loss, but do not exist in the general risk-controlling setting.
>
> It is important to emphasize that utilizing the theory of $\texttt{ACI}$ is not a limitation for a tool intended to solve real-world problems. A crucial consideration in the conformal literature is the design of the score function, where clever suggestions in this regard extend conformal prediction to a broader class of applications, dramatically improving its efficiency and applicability, even though they rely on the existing theory of the conformal framework.
> Our $\texttt{Rolling RC}$ builds on the beautiful and general theory of $\texttt{ACI}$, unlocking new design possibilities and greatly broadening the class of problems that $\texttt{ACI}$ can handle. In addition, we thoroughly study important design choices and propose several stretching functions for $\texttt{Rolling RC}$ that lead to improved performance.
>
> ## ``Online depth estimation use-case is not convincing''
> We thank the reviewer for raising this important point.
> The purpose of the depth estimation example is to demonstrate the flexibility and generality of $\texttt{Rolling RC}$ and to show how our method can be applied to a broader set of problems compared to $\texttt{ACI}$. The problem of online depth estimation from an RGB image is thoroughly investigated in the literature, e.g., [1-6] to name a few contributions. Our $\texttt{Rolling RC}$ can be easily integrated with any of these methods to quantify the uncertainty of the estimated depth.
>
>
> For these reasons, we find ourselves in disagreement with the reviewer's comment that ``Depth estimation use-case is not convincing'', and find $\texttt{Rolling RC}$ as an important contribution for online uncertainty quantification.
> To clarify the significance of online depth estimation in the revised manuscript, we added references to related works on this problem to the introduction.
>
>
>
> ## Online risk-controlling is needed
> Thank you for your advice to discuss in greater depth on the necessity of online risk control. The vanilla $\texttt{ACI}$ can only be applied to control the coverage rate, although in many cases other loss functions are more appropriate. One example of such a task is multi-label classification, where the user may desire to control the fraction of missed classes. Another example is controlling the F1 score which is desired in classification problems. $\texttt{Rolling RC}$ can also control metrics that quantify temporal dependencies, such as the $\texttt{MC}$ metric which serves as a proxy for conditional coverage. In this work, we focused on online depth estimation as use-case because of the extensive interest in this field [1-6]. Another appealing metric to control is the false negative rate in segmentation problems. For instance, [5] finds online image segmentation to be crucial as well. Although the problem of uncertainty quantification for image segmentation was investigated in the offline region [7-9], $\texttt{Rolling RC}$ is currently the only existing method for this task in online settings.
>
> In view of this necessity of our contribution, we ask that the reviewer consider changing their assessment of our work.

---

> ### Author Response · Authors · 2022-12-05
> **Response to the remaining concerns**
>
> Dear reviewer,
> We thank you once again for raising these crucial points.
> 1. We are certain that uncertainty quantification for online depth estimation is highly necessary, as it helps to determine whether the depth estimates are reliable: an algorithm in such an autonomous system may utilize the uncertainty of the depth estimates, provided by $\texttt{Rolling RC}$, when choosing the next action. Therefore, we view online uncertainty quantification as an important tool for decision-making problems that can be used to alert for erroneous predictions or hallucinations.
> 2. We believe that the improved performance of $\texttt{Rolling RC}$ compared to the calibration-set-based method ($\texttt{ACI}$) results from directly updating the interval adjustment parameter (e.g., $\varphi(\theta)$ in equation 6), without relying on a calibration set that may hold outdated samples. Additionally, the careful design choices of the stretching functions that we investigated in this work lead to faster reaction time to changes in the data distribution, improving the performance even further. Figure 7 illustrates that choosing our error-adaptive stretching function results in narrower intervals with better conditional coverage. Importantly, this is one of the main advantages of $\texttt{Rolling RC}$, as it was not proposed in $\texttt{ACI}$ to plug in a stretching function to improve performance. Therefore, while the calibration set size may affect the performance of the calibration-set-based method ($\texttt{ACI}$), we believe that the gain in performance of $\texttt{Rolling RC}$ also arises from the following advantages: (i) directly updating the interval adjustment parameter and (ii) stretching $\theta_t$ with a careful design choice.

---

### Official Review · Reviewer_EbJN · 2022-10-24

**Confidence:** 3
**Correctness:** 4
**Technical Novelty And Significance:** 3
**Empirical Novelty And Significance:** 3
**Recommendation:** 8

**Clarity, Quality, Novelty And Reproducibility:**

This paper is well-written. I enjoy reading this paper and find the presented framework novel and interesting. For clarity, it would be good to have some additional discussions on the theoretical and empirical results - see the questions in the "strength and weaknesses" section.

**Strength And Weaknesses:**

Strength:

1. The proposed method is simple, intuitive, and effective. It has the flexibility to be applied to many different base online learning algorithms with easy implementation.

2. The proposed method can control multiple tasks and provide valid intervals for all requirements over long-range windows in time.

3. Rolling RC is also able to adapt to shifting distributions quickly. The authors also designed several stretching functions for faster adaptation. Their effectiveness is also empirically verified.

Questions and weaknesses:

1. In section 3.2, the authors applied "Rolling RC" to the regression problem by combining the proposed framework with quantile regression. I would assume the proposed framework can also be applied to standard linear regression (i.e., the M_t being the standard linear regression model, instead of a quantile regression model). Is this understanding correct?

2. It seems that the definition of $MC_t$ on page 7 does not match the explanation below. The current definition seems to say $MC_t$ is simply the cumulative miscoverage counter, instead of a counter of "miscoverage events happened in a row".

3. For the experiments in Fig 3, it seems the coverage of energy, traffic, and prices are below 90% when using the "Error" stretching function. How to interpret this result?

4. How to interpret the additional assumption in Thm 3 (as compared to Thm 2)? Why does this new assumption represent that "the risks are more synchronized with each other"?

**Summary Of The Paper:**

In this paper, the authors presented a simple, intuitive, and effective method for quantifying uncertainty for online learning models. The proposed framework provides uncertainty sets that provably control risk, applies to any base online learning algorithm, any user-specified level, and works with non-stationary distributions. The paper presents both empirical and theoretical results that demonstrate the effectiveness of the proposed method.

**Summary Of The Review:**

I enjoy reading this paper and find the results interesting.

---

> ### Author Response · Authors · 2022-11-17
> **Reply to the review**
>
>
> We thank you for the positive feedback and the valuable review. In what follows, we address your concerns in detail.
>
> ## Applying $\texttt{Rolling RC}$ with different learning models $M_t$
> Thank you for bringing up this point. As the reviewer suggests, $\texttt{Rolling RC}$ can be integrated with any online learning method, including linear regression.
>
> ## The definition of the miscoverage counter
> We thank the reviewer for raising this important issue! Indeed, there was a mistake in the definition of the $\texttt{MC}$ metric. The correct definition reads as:
>
> $\texttt{MC}_t =\texttt{MC}_\{t-1\}+1$ if $Y_t \not \in \hat{C}_t(X_t)$, and $0$ otherwise,
>
> where $\texttt{MC}_0=0$.
> We corrected the definition in the main text as well.
>
> ## Coverage below 90\% in Figure 3
> We thank the reviewer for bringing up this point. Indeed, the coverage attained for some of the datasets is not precisely 90\%. However, for all datasets, the coverage rate lies in the interval (89.85, 90.05), which is very close to the nominal 90\% level and aligns with our finite-sample bounds given in Theorem 1. To obtain a coverage rate closer to the nominal level, the user can increase $\gamma$ or enlarge the test set.
>
> ## The additional assumption in Theorem 3
> We thank the reviewer for raising this important subject. The main challenge in exactly controlling multiple risks is satisfying the requirements of all of them simultaneously. That is, it is possible that during the online process one risk will be too high, requiring shrinking the uncertainty set, and the other will be too low, requiring enlarging. Theorem 2 guarantees that all empirical risks are valid by enlarging the prediction sets when at least one risk requires that. However, Theorem 3 states that we can achieve the exact multiple risk control when all the risks are synchronized, that is, one does not require shrinking the prediction set when the other requires enlarging it (and vice versa). In other words, when the risk requirements do not interfere with each other, an exact multiple risk control can be attained.
>
> We thank the reviewer for this feedback, which greatly improves the clarity of our paper!

---

### Official Review · Reviewer_Kmvq · 2022-10-24

**Confidence:** 5
**Clarity, Quality, Novelty And Reproducibility:** See above.
**Correctness:** 4
**Technical Novelty And Significance:** 2
**Empirical Novelty And Significance:** 3
**Recommendation:** 5

**Strength And Weaknesses:**

Strengths:
- Clear and concise abstract and introduction and early problem statement with initial practical example.
- Simple method for controlling multiple risks irrespective of distribution shift.
- Experiments to compare to ACI and practical experiments for image depth prediction.

Weaknesses:
- No separate discussion of related work even though there si quite some work on risk control on conformal prediction for time-series going on.
- In terms of writing, there are also too many references to the appendix in my opinion. This distracts from the main paper. If something is important enough to be discussed in the main paper, please do include at least the gist of it. But, e.g., referring to finite sample bounds and then not providing the exact bounds in the main paper is hindering the reading flow. Many other examples can be found (properly setting \gamma, handling multi-dimensional Y, additional stretching function). I am aware that not everything fits the main paper, but as reader it is difficult to actually decide what is important to read or not.
- Method can be seen as an incremental generalization of ACI, this also concerns the theoretical contributions. Looking at the theorems and their proofs, these seem to follow the ACI paper to most extent – at least I could not spot any difficulties where proofs needed to change or adapt.
- The additional “stretching” functions seem a bit misplaced in the paper. I’d rather have the proofs or more intuition about \gamma, handling multi-dimensional Y, etc. in the main paper). These stretching functions seem more like a “trick” or so – maybe similar to changing conformity scores in conformal prediction.
- Also, regarding the adaptive stretching function, the implications of using examples for this on the guarantee is unclear. This essentially changes the \theta based on the previous examples. But as \theta is already calibrated based on example t – 1, shouldn’t this affect the proof/guarantee somehow?
- In controlling more risks, can I use a single calibration parameter instead of one for each risk? The experiments note that the maximum is used, but in practice I might have parameters independent of the number of risks.
- Proposition 1 seems rather straight-forward, I do not see the need to explicitly highlight this – also considering the proof in the appendix.
- Section A.2 seems just not necessary. I mean the proof is omitted anyway. So why not just have only the more general statement in the main paper with the same/full proof, or just omit it?
- In the experiments, a clear ACI (or even split conformal risk baseline) is missing. As Figure 3 is about coverage, ACI is applicable and should, in my opinion, reduce to ACI – or am I wrong? To be honest, I am not interested in ablating the stretching functions (this is not what the paper is about). So, having proper baselines, maybe showing the time series or having a trivial split baseline seems more interesting.
- Regarding the experiment sin 4.1.3, I am not sure what the takeaway is. I mean for me it seems that this is an empirical strategy to get control over the miscoverage counter. But the method itself cannot provide a guarantee on this, right? Because the guarantee is on losses defined per example.
- The experiments in 4.2 are clearly the main focus of this paper, also given the ages of appendix on this problem and the introductory example. Given that the method is very similar to ACI, I could imagine this paper to put more focus on this paper and concentrate on an audience interested in depth estimation instead of mainly focusing on the guarantees. Currently, the paper is a bit split – regarding reading flow etc. it is hard to really have a consistent story (risk control, stretching functions, depth estimation).
- No conclusion in the end, the paper ends very abruptly.

**Summary Of The Paper:**

The authors propose an online conformal risk control method based on ACI, allowing to control multiple risks irrespective of underlying distribution shifts.

**Summary Of The Review:**

I am not fully convinced that this paper is ready to be published. This is mainly due to a combination of two factors: the story of online (multiple) risk control being diluted by “other things” (stretching functions, the depth estimation application which seems to be key, experiments where I could as well use standard ACI, etc.); and a bit of ambiguity in terms of what the contributions over ACI are (proofs seem to be applicable directly, not a lot of discussion of the theorems etc.). I feel the authors have to decide what kind of paper they want to write and present this in a coherent way, highlighting the corresponding contributions.

---

> ### Author Response · Authors · 2022-11-17
> **Reply to the review (3)**
>
>
> ## Controlling the miscoverage counter
>
> Excellent comment! Indeed, $\texttt{Rolling RC}$ is guaranteed to control losses that are defined over a single sample, and thus cannot be applied to control the $\texttt{MSL}$. To alleviate this, we define the $\texttt{MC}$: a ``sister metric'' to $\texttt{MSL}$ that pushes towards better conditional coverage but is defined over a single sample and therefore is provably controlled by $\texttt{Rolling RC}$. Figure 4 shows that when $\texttt{Rolling RC}$ is applied to control the $\texttt{MC}$, it does rigorously achieve the target $\texttt{MC}$ level while also achieving a valid coverage rate. This stands in contrast with cases where $\texttt{Rolling RC}$ is applied to control the coverage rate alone: it achieves poor $\texttt{MC}$, as indicated by Figure 4c. This experiment demonstrates the great flexibility of $\texttt{Rolling RC}$---it can rigorously control metrics that are defined over the time horizon. We clarified this point in Section 4.1.3 in the revised manuscript.
>
> ## The online depth estimation example
>
> Thank you for raising this subject. The online depth estimation example assists us to showcase the need for risk control and for multiple-risks in online settings. However, online depth estimation is \emph{not} our main goal---we chose this example simply because online depth estimation is a very attractive problem [1-6]. It also demonstrates how the method we offer can elegantly deal with complex scenarios. With that said, we also systematically analyze our $\texttt{Rolling RC}$ with more standard time series data, which is of great importance and value for readers and potential users of this method.
> We updated the introduction section to clarify that the online depth estimation problem is an attractive use-case that $\texttt{ACI}$ cannot handle, while our proposal can. We thank you for helping us clarifying this subject, which greatly improves our paper!
>
> ## No conclusion section
>
> We thank the reviewer for bringing up this comment. Due to space limitations, we could not add a conclusion to the submitted version of the paper. We agree with you that this is an important section and thus added a conclusion to the revised manuscript.
>
> In view of the main purpose of this paper---the proposal of a generic online risk control technique, which is a practical rather than a technical contribution, we politely ask the reviewer to revisit their assessment of our work.
>
> [1] Cho, Younggun, Young-Sik Shin, and Ayoung Kim. "Online depth estimation and application to underwater image dehazing." OCEANS 2016 MTS/IEEE Monterey. IEEE, 2016.
>
> [2] Zhang, Zhenyu, Stephane Lathuiliere, Elisa Ricci, Nicu Sebe, Yan Yan, and Jian Yang. "Online depth learning against forgetting in monocular videos." In Proceedings of the IEEE/CVF Conference on Computer Vision and Pattern Recognition, pp. 4494-4503. 2020.
>
> [3] Maslov, Dmitrii, and Ilya Makarov. "Online supervised attention-based recurrent depth estimation from monocular video." PeerJ Computer Science 6 (2020): e317.
>
> [4] Patil, Vaishakh, Wouter Van Gansbeke, Dengxin Dai, and Luc Van Gool. "Don’t forget the past: Recurrent depth estimation from monocular video." IEEE Robotics and Automation Letters 5, no. 4 (2020): 6813-6820.
>
> [5] Klingner, Marvin, and Tim Fingscheidt. "Online performance prediction of perception DNNs by multi-task learning with depth estimation." IEEE Transactions on Intelligent Transportation Systems 22.7 (2021): 4670-4683.
>
> [6] Jeon, Ikgeun, Liu Yang, Kwangnam Ryu, and Hoon Sohn. "Online melt pool depth estimation during directed energy deposition using coaxial infrared camera, laser line scanner, and artificial neural network." Additive Manufacturing 47 (2021): 102295.

---

> ### Author Response · Authors · 2022-11-17
> **Reply to the review (2)**
>
>
> ## The stretching functions
>
> We thank the reviewer for giving us the opportunity to expand on the stretching functions. $\texttt{Rolling RC}$ unlocks new design possibilities that are not possible in $\texttt{ACI}$, such as the use of stretching functions. The role of the stretching is to adapt faster to changes in the distribution even though $\theta$ is updated only linearly. Indeed, Figure 9 in Appendix D.1.2 shows that applying a stretching function significantly improves performance, and outperforms the calibration-set-based $\texttt{ACI}$ method---achieving faster reaction to distributional shifts. For these reasons, we find the stretching idea to be critical and hence we include it in the main text. Following this discussion, we added a reference to this experiment in the main text in Section 4.1.2.
>
> Regarding the validity of applying a stretching function: to clarify, $\texttt{Rolling RC}$ is guaranteed to be valid independently on the chosen stretching function. Notice that $\varphi$ does not affect $\theta$, but rather the uncertainty set. That is, we calibrate the set using a non-linear function of $\theta$ instead of $\theta$ itself. Nevertheless, if $\theta$ is too small/high (i.e., exceeds the bounds $m,M$), then we still output the empty set/full label set, respectively. This way, the set is guaranteed to achieve small/high risks when necessary in a way that guarantees the risk requirement. We clarified this point in the revised paper in Section 3.2.1.
>
> ## Using a single calibration parameter to control multiple risks
>
> Thank you for bringing up this interesting idea. Indeed, $\texttt{Rolling RC}$ can be applied with calibration parameters that are independent of the number of risks that are controlled. Section 3.3 presents one approach to rigorously control multiple risks, although similar techniques may work as well. We clarified this subject in the last paragraph of Section 3.3.
>
> ## Explicitly stating Proposition 1 in the main text
>
> We thank the reviewer for raising this point. We agree with the reviewer that Proposition 1 is intuitive even though it is stated in the main text. The reason for stating it in the main text is to highlight that controlling $\texttt{MC}$ immediately grants control over the coverage metric. The results are visualized in Figure 4, and are aligned with this statement, indicating that the coverage rate is indeed valid when controlling only the $\texttt{MC}$. Therefore, the goal of Proposition 1 is to show that when applying $\texttt{Rolling RC}$ to control the $\texttt{MC}$ metric, there is no need to explicitly control the coverage rate as well (e.g., with multiple risks control), as the proposition guarantees valid coverage rate.
>
> ## Appendix A.2 - a general version of Theorem 1
>
> Thank you for bringing up this important subject. The importance of the general statement is that $\texttt{Rolling RC}$ can be applied to a broader class of problems, without restricting the prediction set $C$ to be a subset of the full label space. This is useful in multi-label classification tasks, for instance. The role of Appendix A.2 is to show that $\texttt{Rolling RC}$ can be applied in such settings as well. Nevertheless, since this result is not important as the other results in the main text and may distract the reader from our main take-home message, we only briefly explain it in the main text. We find this point important enough to be given in the paper and thus explain it in full detail in the appendix.
>
> ## Comparing $\texttt{Rolling RC}$ to $\texttt{ACI}$
>
> Thank you for this valuable comment which was raised by the other reviewers, and addressed in the reply to all reviewers. For the convenience of the reviewer, we repeat our answer below.
>
> Due to space limitations, we did not include a comparison between $\texttt{Rolling RC}$ and $\texttt{ACI}$ in the main text. Instead, we provided a full comparison of these methods in Figure 9 in Appendix D.1.2, presenting the results of $\texttt{Rolling RC}$ and a scheme that uses a calibration set to construct prediction intervals ($\texttt{ACI}$). This figure shows that $\texttt{Rolling RC}$ applied either without stretching or with stretching performs better than the calibration-set-based $\texttt{ACI}$ method both in terms of conditional coverage and statistical efficiency. Following these results, we believe that $\texttt{Rolling RC}$ is preferable over a vanilla implementation of $\texttt{ACI}$. In response, we added a reference to this comparison in the main text in Section 4.1.2.

---

> ### Author Response · Authors · 2022-11-17
> **Reply to the review (1)**
>
>
> We thank you for the positive feedback, suggestions, and encouragement. In what follows, we address your concerns in detail.
>
> ## Related work discussion
> We thank the reviewer for raising this important point. In Section 2 we discussed existing methods for risk-controlling in offline settings and online methods for controlling the coverage rate. We are not aware of existing methods for risk control in online settings other than $\texttt{Rolling RC}$. We are only aware of conformal prediction methods for time series that target coverage/miscoverage (i.e., a binary loss). We would appreciate it if you could share with us other techniques for this task. We agree with you that describing related work is crucial, so we will add a discussion about the techniques you share with us to the main text.
>
> We thank the reviewer for this comment, which greatly improves our paper!
>
>
> ## Referring to the appendix.
>
> Thank you for bringing up this important subject. Indeed, there are many references to the appendix from the main text due to space limitations. For this reason, we included in the main text information we view as the most important---that $\texttt{Rolling RC}$ can be applied to control a general risk and even control multiple risks at once. We referred to the appendix only in cases where the subject distracts the user from our main take-home message, such as properly choosing $\gamma$, technical details on depth estimation, or additional weaker stretching functions.
>
> In response, we added more information to the main text, such as the finite-sample bounds given by Theorem 1, and reduced the number of references to the appendix, to keep the flow as fluent as possible. Thank you for helping us to improve our writing!
>
>
> ## The novelty of $\texttt{Rolling RC}$
>
> Thank you for this valuable comment which was raised by the other reviewers, and addressed in the reply to all reviewers. In short, we agree that we are not making a mathematical contribution about online learning, but rather a practical one. A reader of $\texttt{ACI}$ would not know how to apply it to control an arbitrary risk. In the paper, we propose a general framework to control an arbitrary risk, which we view as a useful and insightful contribution for readers who need an algorithm for uncertainty quantification in the online setting. We updated the introduction to clarify this point. For the convenience of the reviewer, we repeat the full answer below.
>
> We agree that from a theoretical perspective we rely on the mathematical foundations of $\texttt{ACI}$, but we do not see this as a limitation, as our contribution is practical rather than technical. In this work, we show how to broaden the set of problems that $\texttt{ACI}$ can tackle and control an arbitrary loss, which is far from being straightforward. To put it simply, a reader of $\texttt{ACI}$ would not know how to use this technique in online risk-controlling tasks, such as the depth sensing example. Specifically, one cannot naively plug an arbitrary loss function into $\texttt{ACI}$ to achieve risk control. The reason is that $\texttt{ACI}$ works with conformity scores that are relevant only to the miscoverage loss, but do not exist in the general risk-controlling setting.
>
> It is important to emphasize that utilizing the theory of $\texttt{ACI}$ is not a limitation for a tool intended to solve real-world problems. A crucial consideration in the conformal literature is the design of the score function, where clever suggestions in this regard extend conformal prediction to a broader class of applications, dramatically improving its efficiency and applicability, even though they rely on the existing theory of the conformal framework.
> Our $\texttt{Rolling RC}$ builds on the beautiful and general theory of $\texttt{ACI}$, unlocking new design possibilities and greatly broadening the class of problems that $\texttt{ACI}$ can handle. In addition, we thoroughly study important design choices and propose several stretching functions for $\texttt{Rolling RC}$ that lead to improved performance.

---

> ### Author Response · Authors · 2022-12-02
> **Happy to provide additional clarification**
>
> We would like to thank you once again for your thorough review, many helpful comments, and the interesting questions raised. We would like to ask you to, please, review our responses, and consider reassessing our work. If you have any questions, comments, or concerns left, please, let us know, we are eager to engage in further discussion.

---

### Official Review · Reviewer_9cEN · 2022-10-24

**Confidence:** 3
**Correctness:** 4
**Technical Novelty And Significance:** 1
**Empirical Novelty And Significance:** Not applicable
**Recommendation:** 3

**Clarity, Quality, Novelty And Reproducibility:**

I think the paper is well-written and easy to read. In that sense, it's very easy for the readers to get the points that the author/s are delivering. But I have some doubts for the novelty of the work as I commented above as the weakness of the paper.

**Strength And Weaknesses:**

Strength:
  1. The framework for constructing uncertainty sets has theoretical guarantee on the risk control, which is also extended to control multiple risks simultaneously.
  2. The theoretical result is applicable even when the underlying data has drastic distribution shifts over time in an unknown manner.

Weakness:
  The results are very incremental to the previous works such as Gibbs & Candes, 2021 for the theoretical findings and Angelopoulos et al., 2021a; 2022a; Bates et al.,2021 for the different control metrics extension besides coverage rate

**Summary Of The Paper:**

This paper proposes a framework to construct uncertainty sets in the online setting to control the risk of the coverage, false negative rate, or F1 score. The propose method has theoretical guarantee for risk control at the user specified level for different underlying data like distribution shifts over time in an unknown manner. It is also extended to control multiple risks simultaneously and experiments show the proposed method rigorously controls various natural risks.


**Summary Of The Review:**

I think the problem the paper is trying to solve is very interesting and meaningful. The paper is well-written and easy to read, which is good for the readers to learn the main points of the paper. Although the theoretical results look stronger than previous work in the online learning setting, the technics it uses are very incremental and would not provide much insights for the potential readers.

---

> ### Author Response · Authors · 2022-11-17
> **Response to the review**
>
> We appreciate your review of our work and thank you for the helpful suggestions and interest in our work. In what follows we respond to your concerns in detail.
>
> ## The novelty of $\texttt{Rolling RC}$
>
> The reviewer's main concern seems to be that ``the techniques it uses are very incremental and would not provide much insights for the potential readers.'' We agree that we are not making a mathematical contribution about online learning, but rather a practical one. A reader of $\texttt{ACI}$ would not know how to apply it to control an arbitrary risk, as required, for instance, in the depth sensing example described in the paper. Nevertheless, the method we propose in this work allows the user to intuitively control any given risk, which we view as useful and insightful for readers who need an algorithm for uncertainty quantification in the online setting.
> For the convenience of the reviewer, below we repeat the answer given in the reply to all reviewers.
>
> We agree that from a theoretical perspective we rely on the mathematical foundations of $\texttt{ACI}$, but we do not see this as a limitation, as our contribution is practical rather than technical. In this work, we show how to broaden the set of problems that $\texttt{ACI}$ can tackle and control an arbitrary loss, which is far from being straightforward. To put it simply, a reader of $\texttt{ACI}$ would not know how to use this technique in online risk-controlling tasks, such as the depth sensing example. Specifically, one cannot naively plug an arbitrary loss function into $\texttt{ACI}$ to achieve risk control. The reason is that $\texttt{ACI}$ works with conformity scores that are relevant only to the miscoverage loss, but do not exist in the general risk-controlling setting.
>
> It is important to emphasize that utilizing the theory of $\texttt{ACI}$ is not a limitation for a tool intended to solve real-world problems. A crucial consideration in the conformal literature is the design of the score function, where clever suggestions in this regard extend conformal prediction to a broader class of applications, dramatically improving its efficiency and applicability, even though they rely on the existing theory of the conformal framework.
> Our $\texttt{Rolling RC}$ builds on the beautiful and general theory of $\texttt{ACI}$, unlocking new design possibilities and greatly broadening the class of problems that $\texttt{ACI}$ can handle. In addition, we thoroughly study important design choices and propose several stretching functions for $\texttt{Rolling RC}$ that lead to improved performance.
>
> ## Previous works on risk control
> We thank the reviewer for raising this important point. The work on risk control in offline settings serves as a motivation for our proposal, emphasizing its necessity. Nevertheless, the approach we offer is fundamentally different: we rely on a different set of mathematical and statistical ideas. That is, although both $\texttt{Rolling RC}$ and conformal risk-controlling approaches have a risk control guarantee, they accomplish this goal in different ways. For instance, one major difference is that offline risk control techniques use a holdout set to calibrate the prediction intervals, while $\texttt{Rolling RC}$ does not. Another difference is that $\texttt{Rolling RC}$ updates the calibration parameter $\theta_t$ online, while offline risk-controlling methods set their calibration parameter only once.
>
> We politely ask the reviewer to revisit their assessment of our work, in view of the practical rather than technical nature of the contribution we are making.

---

> ### Author Response · Authors · 2022-12-02
> **Happy to provide additional clarification**
>
> We would like to thank you once again for your thorough review, many helpful comments, and the interesting questions raised. We would like to ask you to, please, review our responses, and consider reassessing our work. If you have any questions, comments, or concerns left, please, let us know, we are eager to engage in further discussion.

---

### Author Response · Authors · 2022-11-17
**Response to all reviews**

Dear Reviewers,
Thank you for your time and effort in reviewing our submission and providing valuable feedback and suggestions. We are excited to see that the reviewers found our proposal valuable; to quote Reviewer EbJN:
> The proposed method is simple, intuitive, and effective. It has the flexibility to be applied to many different base online learning algorithms with easy implementation.

and also Reviewer Kmvq:
> Simple method for controlling multiple risks irrespective of distribution shift.


In addition, the reviewers found the fact that our online calibration scheme can control an arbitrary risk in online adversarial settings is novel; to quote Reviewer sghz:
> $\texttt{Rolling RC}$ can control any risk, which is novel in online settings.

Below, we briefly summarize the major comments raised, where we provide detailed answers to all reviewers’ concerns in the response to each review.

The main concern raised by the reviewers questions the novelty of $\texttt{Rolling RC}$. We agree that from a theoretical perspective we rely on the mathematical foundations of $\texttt{ACI}$, but we do not see this as a limitation, as our contribution is practical rather than technical. In this work, we show how to broaden the set of problems that $\texttt{ACI}$ can tackle and control an arbitrary loss, which is far from being straightforward. To put it simply, a reader of $\texttt{ACI}$ would not know how to use this technique in online risk controlling tasks, such as the depth sensing example. Specifically, one cannot naively plug an arbitrary loss function into $\texttt{ACI}$ to achieve risk control. The reason is that $\texttt{ACI}$ works with conformity scores that are relevant only to the miscoverage loss, but do not exist in the general risk controlling-setting.

It is important to emphasize that utilizing the theory of $\texttt{ACI}$ is not a limitation for a tool intended to solve real-world problems. A crucial consideration in the conformal literature is the design of the score function, where clever suggestions in this regard extend conformal prediction to a broader class of applications, dramatically improving its efficiency and applicability, even though they rely on the existing theory of the conformal framework.
Our $\texttt{Rolling RC}$ builds on the beautiful and general theory of $\texttt{ACI}$, unlocking new design possibilities and greatly broadening the class of problems that $\texttt{ACI}$ can handle. In addition, we thoroughly study important design choices and propose several stretching functions for $\texttt{Rolling RC}$ that lead to improved performance, as explained hereafter.


Reviewer Kmvq and Reviewer sghz asked for a comparison between $\texttt{Rolling RC}$ and $\texttt{ACI}$. Due to space limitations, we did not include this comparison in the main text. Instead, we provided a full comparison of these methods in Figure 9 in Appendix D.1.2, presenting the results of $\texttt{Rolling RC}$ and a scheme that uses a calibration set to construct prediction intervals ($\texttt{ACI}$). This figure shows that $\texttt{Rolling RC}$ applied either without stretching or with stretching performs better than the calibration-set-based $\texttt{ACI}$ method both in terms of conditional coverage and statistical efficiency. Following these results, we believe that $\texttt{Rolling RC}$ is preferable over a vanilla implementation of $\texttt{ACI}$. In response, we added a reference to this comparison in the main text in Section 4.1.2.

For these reasons we find $\texttt{Rolling RC}$ as an important contribution for online uncertainty quantification and believe it will be very useful for many researchers across the world.

---

### Decision · Program_Chairs · 2023-01-20

**Decision:**

Reject

**Justification For Why Not Higher Score:**

See the above.

**Justification For Why Not Lower Score:**

N/A

**Metareview: Summary, Strengths And Weaknesses:**

The paper proposes a conformal prediction-style uncertainty quantification method for online learning.

The strengths are numerous:
-  The algorithm is simple and effective --- guaranteed to produce the right coverage.
- works with any online learner as a black box.
-  it controls many popular risk measures irrespective of underlying distribution shifts.

The main weakness that reviewers highlighted is the weak technical novelty over existing work.  Reviewers also found the empirical evaluation to be unconvincing.

**Summary Of Ac-Reviewer Meeting:**

This paper is on the borderline.  Reviewers and AC discussed the pros and cons extensively.  Despite the differences in the scores, most reviewers agree on the assessment that the technical novelty of the work is low relative to ACI and other related work.  The authors argued that the merits of the work lie primarily in the practical aspects. However, reviewers found that the experiments in the paper are not thoroughly developed for the paper to be accepted based on experiments alone.